# Collaborative Algorithms for Online Personalized Mean Estimation

**Mahsa Asadi**                                                                 *llvllahsa@gmail.com*
*Univ. Lille, Inria, CNRS, Centrale Lille*
*UMR 9189 - CRIStAL, F-59000 Lille, France*

**Aurélien Bellet**                                                         *aurelien.bellet@inria.fr*
*Univ. Lille, Inria, CNRS, Centrale Lille*
*UMR 9189 - CRIStAL, F-59000 Lille, France*

**Odalric-Ambrym Maillard**                                             *odalric.maillard@inria.fr*
*Univ. Lille, Inria, CNRS, Centrale Lille*
*UMR 9189 - CRIStAL, F-59000 Lille, France*

**Marc Tommasi**                                                          *marc.tommasi@inria.fr*
*Univ. Lille, Inria, CNRS, Centrale Lille*
*UMR 9189 - CRIStAL, F-59000 Lille, France*

**Reviewed on OpenReview:** *https://openreview.net/forum?id=VipljNfZSZ*

## Abstract

We consider an online estimation problem involving a set of agents. Each agent has access to a (personal) process that generates samples from a real-valued distribution and seeks to estimate its mean. We study the case where some of the distributions have the same mean, and the agents are allowed to actively query information from other agents. The goal is to design an algorithm that enables each agent to improve its mean estimate thanks to communication with other agents. The means as well as the number of distributions with same mean are unknown, which makes the task nontrivial. We introduce a novel collaborative strategy to solve this online personalized mean estimation problem. We analyze its time complexity and introduce variants that enjoy good performance in numerical experiments. We also extend our approach to the setting where clusters of agents with similar means seek to estimate the mean of their cluster.

## 1 Introduction

With the widespread of personal digital devices, ubiquitous computing and IoT (Internet of Things), the need for decentralized and collaborative computing has become more pressing. Indeed, devices are first of all designed to collect data and this data may be sensitive and/or too large to be transmitted. Therefore, it is often preferable to keep the data on-device, where it has been collected. Local processing on a single device is a always possible option but learning in isolation suffers from slow convergence time when data arrives slowly. In that case, collaborative strategies can be investigated in order to increase statistical power and accelerate learning. In recent years, such collaborative approaches have been broadly referred to as federated learning (Kairouz et al., 2021).

The data collected at each device reflects the specific usage, production patterns and objective of the associated agent. Therefore, we must solve a set of *personalized tasks over heterogeneous data distributions*. Even though the tasks are personalized, collaboration can play a significant role in reducing the time complexity and accelerating learning in presence of agents who share similar objectives. An important building block to design collaborative algorithms is then to identify agents acquiring data from the same (or similar) distri-

bution. This is particularly difficult to do in an *online* setting, in which data becomes available sequentially over time.

In this work, we explore this challenging objective in the context of a new problem: *online personalized mean estimation.* Formally, each agent continuously receives data from a *personal $\sigma$-sub-Gaussian distribution* and aims to construct an accurate estimation of its mean as fast as possible. At each step, each agent receives a new sample from its distribution but is also allowed to query the current local average of another agent. To enable collaboration, we assume the existence of an underlying *class structure* where agents in the same class have the same mean value. We also consider a relaxed assumption where the means of agents in a class are close (but not necessarily equal). Such assumptions are natural in many real-world applications (Adi et al., 2020). A simple example is that of in different environments, monitoring parameters such as temperature in order to accurately estimate their mean (see for instance Mateo et al., 2013). Another example is collaborative filtering, where the goal is to estimate user preferences by leveraging the existence of clusters of users with similar preferences (Su & Khoshgoftaar, 2009a). Crucially, the number of classes and their cardinality are unknown to the agents and must be discovered in an online fashion.

We propose collaborative algorithms to solve this problem, where agents identify the class they belong to in an online fashion so as to better and faster estimate their own mean by assigning weights to other agents' estimates. Our approach is grounded in Probably Approximately Correct (PAC) theory, allowing agents to iteratively discard agents in different classes with high confidence. We provide a theoretical analysis of our approach by bounding the time required by an agent to correctly estimate its class with high probability, as well as the time required by an agent to estimate its mean to the desired accuracy. Our results highlight the dependence on the gaps between the true means of agents in different classes, and show that in some settings our approach achieves nearly the same time complexity as an oracle who would know the classes beforehand. Our numerical experiments on synthetic data are in line with our theoretical findings and show that some empirical variants of our approach can further improve the performance in practice.

The paper is organized as follows. Section 2 discusses the related work on federated learning and collaborative online learning. In Section 3, we formally describe the problem setting and introduce relevant notations. In Section 4, we introduce our algorithm and its variants. Section 5 presents our theoretical analysis of the proposed algorithm in terms of class and mean estimation time complexity. Section 6 is devoted to illustrative numerical experiments. Section 7 extends our approach to the case where classes consist of agents with similar (but not necessarily equal) means and agents seek to estimate the mean of their class. We conclude and discuss perspectives for future work in Section 8.

## 2    Related Work

Over the last few years, collaborative estimation and learning problems involving several agents with local datasets have been extensively investigated under the broad term of Federated Learning (FL) (Kairouz et al., 2021). While traditional FL algorithms learn a global estimate for all agents, more personalized approaches have recently attracted a lot of interest (see for instance Vanhaesebrouck et al., 2017; Smith et al., 2017; Fallah et al., 2020; Sattler et al., 2020; Hanzely et al., 2020; Marfoq et al., 2021, and references therein). With the exception of recent work on simple linear regression settings (Cheng et al., 2021), these approaches typically lack clear statistical assumptions on the relation between local data distributions and do not provide error guarantees with respect to these underlying distributions. More importantly, the above methods focus on the batch learning setting and are not suitable for online learning.

In the online setting, the work on collaborative learning has largely focused on multi-armed bandits (MAB). Most approaches however consider a *single* MAB instance which is solved collaboratively by multiple agents. Collaboration between agents can be implemented through broadcast messages to all agents (Hillel et al., 2013; Tao et al., 2019), via a server (Wang et al., 2020b), or relying only on local message exchanges over a network graph (Sankararaman et al., 2019; Martínez-Rubio et al., 2019; Wang et al., 2020a; Landgren et al., 2021; Madhushani et al., 2021). Other approaches do not allow explicit communication but instead consider a collision model where agents receive no reward if several agents pull the same arm (Boursier & Perchet, 2019; Wang et al., 2020a). In any case, all agents aim at solving the *same* task.

Some recent work considered collaborative MAB settings where the arm means vary across agents. Extending their previous work (Boursier & Perchet, 2019), Boursier et al. (2020) consider the case where arm means can vary among players. Under their collision model, the problem reduces to finding a one-to-one assignment of agents to arms. In Shi & Shen (2021), the local arm means of each agent are IID random realizations of fixed global means and the goal is to solve the global MAB using only observations from the local arms with an algorithm inspired from traditional FL. Similarly, Karpov & Zhang (2022) extend the work of Tao et al. (2019) by considering different local arm means for each agent with the goal to identify the arm with largest aggregated mean. Shi et al. (2021) introduce a limited amount of personalization by extending the model of Shi & Shen (2021) to optimize a mixture between the global and local MAB objectives. Réda et al. (2022) further consider a *known* weighted combination of the local MAB objectives, and focus on the pure exploration setting (best arm identification) rather than regret minimization. A crucial difference with our work is that there is no need to discover relations between local distributions to solve the above problems.

Another related problem is to identify a graph structure on top of the arms in MAB. Kocák & Garivier (2020; 2021) construct a similarity graph while solving the best arm identification problem, but consider only a single agent. In contrast, our work considers a multi-agent setting with personalized estimation tasks, and our approach discovers similarities across agents' tasks in an online manner.

## 3 Problem Setting

We consider a mean estimation problem involving $A$ agents. The goal of each agent $a \in [A] = \{1, 2, \ldots, A\}$ is to estimate the mean $\mu_a$ of a personal distribution $\nu_a$ over $\mathbb{R}$. In this work, we assume that there exists $\sigma \geq 0$ such that each $\nu_a$ is $\sigma$-sub-Gaussian, i.e.:

$$\forall \lambda \in \mathbb{R}, \quad \log \mathbb{E}_{x \sim \nu_a} \exp(\lambda(x - \mu_a)) \leq \frac{1}{2}\lambda^2\sigma^2.$$

This classical assumption captures a property of strong tail decay, and includes in particular Gaussian distributions (in that case, the smallest possible $\sigma^2$ corresponds to the variance) as well as any distribution supported on a bounded interval (e.g., Bernoulli distributions).

We consider an online and collaborative setting where data points are received sequentially and agents can query each other to learn about their respective distributions. Agents should be thought of as different user devices which operate in parallel. Therefore, they all receive a new sample and query another agent at each time step.

Formally, we assume that time is synchronized between agents and at each time step $t$, each agent $a$ receives a new sample $x_a^t$ from its personal distribution $\nu_a$ with mean $\mu_a$, which is used to update its local mean estimate $\bar{x}_{a,a}^t = \frac{1}{t}\sum_{t'=1}^{t} x_a^{t'}$. It also chooses another agent $l$ to *query*. As a response from querying agent $l$, agent $a$ receives the local average $\bar{x}_{l,l}^t$ of agent $l$ (i.e., the average of $t$ independent samples from the personal distribution $\nu_l$) and stores it in its memory $\bar{x}_{a,l}^t$ along with the corresponding number of samples $n_{a,l}^t = t$. Each agent $a$ thus keeps a memory $[(\bar{x}_{a,1}^t, n_{a,1}^t), \ldots, (\bar{x}_{a,A}^t, n_{a,A}^t)]$ of the last local averages (and associated number of samples) that it received from other agents. The information contained in this memory is used to compute an *estimate* $\mu_a^t$ of $\mu_a$ at each time $t$. Our goal is to design a query and estimation procedure for each agent.

As described above, note that when an agent queries another agent at time $t$, it does not receive one sample from this agent (as e.g. in multi-armed bandits), but receives the full statistics of observations of this agent up to time $t$. This is considerably much more information than in typical MAB settings, and naturally requires specific strategies.

The goal of each agent to find a good estimate of its personal mean as fast as possible, without consideration for the quality of the estimates in earlier steps (i.e., we do not seek to minimize a notion of regret). In the online learning terminology, this is referred to as a pure exploration setting (like best arm identification in multi-armed bandits, see Audibert et al., 2010). Formally, we will measure the performance of an algorithm using the notion of $(\epsilon, \delta)$-convergence in probability (Bertsekas & Tsitsiklis, 2002; Wasserman, 2013), which we recall below.

**Definition 1** (PAC-convergence). *An estimation procedure for agent $a$ is called $(\epsilon, \delta)$-convergent if there exists $\tau_a \in \mathbb{N}$ such that the probability that the mean estimator $\mu_a^t$ of agent $a$ is $\epsilon$-distant from the true mean for any time $t > \tau_a$ is at least $1 - \delta$:*

$$\mathbb{P}\left(\forall t > \tau_a : |\mu_a^t - \mu_a| \leq \epsilon\right) > 1 - \delta. \tag{1}$$

While it is easy to design $(\epsilon, \delta)$-convergent estimation procedure for a single agent taken in isolation, the goal of this paper is to propose collaborative algorithms where agents benefit from information from other agents by taking advantage of the relation between the agents' distributions. This will allow them to build up more accurate estimations in less time, i.e., with smaller time complexity $\tau_a$.

Specifically, to foster collaboration between agents, we consider that the set of agents $[A]$ is partitioned into equivalence classes that correspond to agents with the same mean.[1] In real scenarios, these classes may represent sensors in the same environment, objects with the same technical characteristics, users with the same behavior, etc. This assumption makes it possible for an agent to design strategies to identify other agents in the same class and to use their estimates in order to speed up the estimation of his/her own mean. Formally, we define the class of $a$ as the set of agents who have the same mean as $a$.

**Definition 2** (Similarity class). *The* similarity class *of agent $a$ is given by:*

$$\mathcal{C}_a = \{l \in [A] : \Delta_{a,l} = 0\},$$

*where $\Delta_{a,l} = |\mu_a - \mu_l|$ is the gap between the means of agent $a$ and agent $l$.*

The gaps $\{\Delta_{a,l}\}_{a,l \in [A]}$ define the problem structure. We consider that the agents do not know the means, the gaps, or even the number of underlying classes. Hence the classes $\{\mathcal{C}_a\}_{a \in [A]}$ are completely unknown. This makes the problem quite challenging.

**Remark 1** (Scalability). *In large-scale systems, it may be impractical for agents to query all other agents and/or to maintain a memory size that is linear in the number of agents $A$. From a practical point of view, each agent can instead consider a restricted subset of agents of reasonable size. This subset could be picked uniformly at random, be composed of neighboring nodes in the network or in the physical world, or be based on prior knowledge on who is more likely to be in the same class (when available).*

## 4 Proposed Approach

In this section, we first introduce some of the key technical components used in our approach, and then present our proposed algorithm.

### 4.1 Main Concepts

In our approach, each agent $a$ computes confidence intervals $I_{a,l} = [\bar{x}_{a,l}^t - \beta_\delta(n_{a,l}^t), \bar{x}_{a,l}^t + \beta_\delta(n_{a,l}^t)]$ for the mean estimators $[\bar{x}_{a,1}^t, \ldots, \bar{x}_{a,A}^t]$ that it holds in memory at time $t$. The generic confidence bound $\beta_\delta(n_{a,l}^t)$ takes as input the number of samples $n_{a,l}^t$ seen for agent $l$ at time $t$, and $\delta$ corresponds to the risk parameter so that with probability at least $1 - \delta$ the true mean $\mu_l$ falls within the confidence interval $I_{a,l}$.

Agent $a$ will use these confidence intervals to assess whether another agent $l$ belongs to the class $\mathcal{C}_a$ through the evaluation of an optimistic distance defined below.

**Definition 3** (Optimistic distance). *The* optimistic distance *with agent $l$ from the perspective of agent $a$ is defined as:*

$$d_{a,\delta}^t(l) = |\bar{x}_{a,a}^t - \bar{x}_{a,l}^t| - \beta_\delta(n_{a,a}^t) - \beta_\delta(n_{a,l}^t). \tag{2}$$

The "optimistic" terminology is justified by the fact that $d_{a,\delta}^t(l)$ is, with high probability, a *lower bound* on the distance between the true means $\mu_a$ and $\mu_l$ of distributions $\nu_a$ and $\nu_l$. Recall that two agents belong to

---

[1]In Section 7, we will consider a relaxed version of this assumption where classes consist of agents with *similar* (not necessarily equal) means.

---

**Algorithm 1** *ColME*

---

**Parameters:** agent $a$, time horizon $H$, risk $\delta$, weighting scheme $\alpha$, and query strategy *choose_agent*

$\quad \forall l \in [A]: \bar{x}^0_{a,l} \leftarrow 0, n^0_{a,l} \leftarrow 0$

$\quad \mathcal{C}^0_a \leftarrow \{l \in [A] : d^t_{a,\delta}(l) \leq 0\} = [A]$

$\quad$ **for** $t = 1, \ldots, H$ **do**

$\qquad \forall l \in [A]: \bar{x}^t_{a,l} \leftarrow \bar{x}^{t-1}_{a,l}, n^t_{a,l} \leftarrow n^{t-1}_{a,l}$

$\qquad$ **Perceive:**

$\qquad\quad$ Receive sample $x^t_a \sim \nu_a$

$\qquad\quad \bar{x}^t_{a,a} \leftarrow \bar{x}^{t-1}_{a,a} \times \frac{t-1}{t} + x^t_a \times \frac{1}{t}, \;\; n^t_{a,a} \leftarrow t$

$\qquad$ **Query:**

$\qquad\quad \mathcal{C}^t_a \leftarrow \{l \in [A] : d^t_{a,\delta}(l) \leq 0\}$

$\qquad\quad$ Query agent $l = choose\_agent(\mathcal{C}^t_a)$ to get $\bar{x}^t_{l,l}$

$\qquad\quad \bar{x}^t_{a,l} \leftarrow \bar{x}^t_{l,l}, \;\; n^t_{a,l} \leftarrow t$

$\qquad$ **Estimate:**

$\qquad\quad \mathcal{C}^t_a \leftarrow \{l \in [A] : d^t_{a,\delta}(l) \leq 0\}$

$\qquad\quad \mu^t_a \leftarrow \sum_{l \in \mathcal{C}^t_a} \alpha^t_{a,l} \times \bar{x}^t_{a,l}$

$\quad$ **end for**

**Output:** $\mu^H_a$

---

the same class if the distance of their true mean is zero. Since these values are unknown, the idea of the above heuristic is to provide a proxy based on observed data and high probability confidence bounds. In particular, we will adopt the *Optimism in Face of Uncertainty Principle (OFU)* (see Auer et al., 2002) and consider that two agents may be in the same class if the optimistic distance is zero or less. Hence, we define an optimistic notion of class accordingly.

**Definition 4.** *The* optimistic similarity class *from the perspective of agent $a$ at time $t$ is defined as:*

$$\mathcal{C}^t_a = \{l \in [A] : d^t_{a,\delta}(l) \leq 0\}.$$

Having introduced the above concepts, we can now present our algorithm.

### 4.2 Algorithm

The collaborative mean estimation algorithm we propose, called *ColME*, is given in Algorithm 1 (taking the perspective of agent $a$). For conciseness, we consider that $\beta_\delta(0) = +\infty$. At each step $t$, agent $a$ performs three main steps.

In the **Perceive** step, the agent receives a sample from its distribution and updates its local average together with the number of samples.

In the **Query** step, agent $a$ selects another agent following a query strategy given as a parameter to the *ColME* algorithm. Agent $a$ runs the *choose_agent* function to select another agent $l$ and asks for its current local estimate to update its memory. We propose two variants for the *choose_agent* function:

- *Round-Robin*: cycle over the set $[A]$ of agents one by one in a fixed order.

- *Restricted-Round-Robin*: like round-robin but ignores agents that are not in the set of optimistically similar agents $\mathcal{C}^t_a$.

The focus on round-robin-style strategies is justified by the information structure of our problem setting, which is very different from classic bandits. Indeed, querying an agent at time $t$ produces an estimate computed on the $t$ observations collected by this agent so far. The choice of variant (*Round-Robin* or *Restricted-Round-Robin*) will affect the class identification time complexity, as we shall discuss later.

Finally, in the **Estimate** step, agent $a$ computes the optimistic similarity class $\mathcal{C}_a^t$ based on available information, and constructs its mean estimate as a weighted aggregation of the local averages of agents that belong to $\mathcal{C}_a^t$. We propose different weighting mechanisms:

**Simple weighting.**   This is a natural weighting mechanism for aggregating samples:

$$\alpha_{a,l}^t = \frac{n_{a,l}^t}{\sum_{l \in C_a^t} n_{a,l}^t}.$$

**Soft weighting.**   This is a heuristic weighting mechanism which leverages the intuition that the more the confidence intervals of two agents overlap, the more likely that they are in the same class. Moreover, the smaller the union of the agent means, the more confident we are that the agents are in the same class. In other words, we are not equally confident about all the agents that are selected for estimation, and this weighting mechanism incorporates this information:

$$\alpha_{a,l}^t = n_{a,l}^t \frac{|I_{a,a} \cap I_{a,l}|}{|I_{a,a} \cup I_{a,l}|} \times \frac{1}{Z_{\text{soft}}},$$

where $Z_{\text{soft}} = \sum_{i \in C_a^t} \frac{n_{a,i}^t |I_{a,a} \cup I_{a,i}|}{|I_{a,a} \cap I_{a,i}|}$ is a normalization factor.

**Aggressive weighting.**   This is an extension of the previous soft weighting mechanism that is more selective. Not only does it consider the overlap and intersection of the agents' confidence intervals, but it also requires the size of the intersection to be larger than half the size of both confidence intervals from the two agents. Let us denote the binary value associated with this condition by $E_{a,l} = \mathbb{1}_{\{|I_{a,a} \cap I_{a,l}| > \min\{\beta_\delta(n_{a,l}^t), \beta_\delta(n_{a,a}^t)\}\}}$. Then

$$\alpha_{a,l}^t = n_{a,l}^t \frac{|I_{a,a} \cap I_{a,l}|}{|I_{a,a} \cup I_{a,l}|} \times \frac{E_{a,l}}{Z_{\text{agg}}},$$

where $Z_{\text{agg}} = \sum_{i \in C_a^t} \frac{n_{a,i}^t |I_{a,a} \cup I_{a,i}| \times E_{a,i}}{|I_{a,a} \cap I_{a,i}|}$ is a normalization factor.

### 4.3   Baselines

We introduce two baselines that will be used to put the performance of our approach into perspective, both theoretically and empirically.

**Local estimation.**   Estimates are computed without any collaboration, using only samples received from the agent's own distribution, i.e. $\mu_a^t = \bar{x}_{a,a}^t$.

**Oracle weighting.**   The agent knows the true class $\mathcal{C}_a$ via an oracle and uses the simple weighting $\alpha_{a,l}^t = \frac{n_{a,l}^t}{\sum_{l \in \mathcal{C}_a} n_{a,l}^t}$.

## 5   Theoretical Analysis

In this section, we provide a theoretical analysis of our algorithm *ColME* for the query strategy *Restricted-Round-Robin* and the simple weighting scheme. Specifically, we bound the time complexity in probability for both class and mean estimation. Below, we explain the key steps involved in our analysis and state our main results. All proofs can be found in the appendix.

A key aspect of our analysis is to characterize when the optimistic similarity class (Definition 4) coincides with the true classes. We show that this is the case when two conditions hold. First, for a given agent $a$, we need the confidence interval computed by $a$ about agent $l$ to contain the true mean $\mu_l$ for all $l \in A$.

**Definition 5.** *We define the following events:*

$$E_a^t = \bigcap_{l \in [A]} |\bar{x}_{a,l}^t - \mu_l| \le \beta_\delta(n_{a,l}^t), \tag{3}$$

$$E_a = \bigcap_{t \in \mathbb{N}} E_a^t. \tag{4}$$

We can guarantee that $E_a$ holds with high probability via an appropriate parameterization of confidence intervals. We use the so-called Laplace method (Maillard, 2019).

**Lemma 1.** *Let $\delta \in (0,1)$, $a \in [A]$. Setting $\beta_\delta(n) = \sigma\sqrt{2\frac{1}{n} \times (1 + \frac{1}{n})\ln(\sqrt{n+1}/\gamma(\delta))}$ with $\gamma(\delta) = \frac{\delta}{8 \times A}$, we have:*

$$\mathbb{P}(E_a) \ge 1 - \frac{\delta}{8}. \tag{5}$$

The second condition is that agent's $a$ memory about the local estimates of other agents should contain enough samples. Let us denote by $\lceil \beta_\delta^{-1}(x) \rceil$ the smallest integer $n$ such that $x > \beta_\delta(n)$.

**Definition 6.** *From the perspective of agent $a$ and at time $t$, event $G_a^t$ is defined as:*

$$G_a^t = \bigcap_{l \in [A]} n_{a,l}^t > n_{a,l}^\star, \tag{6}$$

$$\text{where} \qquad n_{a,l}^\star = \begin{cases} \lceil \beta_\delta^{-1}(\frac{\Delta_{a,l}}{4}) \rceil & \text{if } l \notin \mathcal{C}_a, \\ \lceil \beta_\delta^{-1}(\frac{\Delta_a}{4}) \rceil & \text{otherwise,} \end{cases}$$

*with $\Delta_a = \min_{l \in [A] \backslash \mathcal{C}_a} \Delta_{a,l}$.*

Note that the required number of samples is inversely proportional to the gaps between the means of agents in different classes. Having enough samples and knowing that the true means fall within the confidence bounds, we can show that the class-estimation rule $d_{a,\delta}^t(l) \le 0$ indicates the membership of $l$ in $\mathcal{C}_a$.

**Lemma 2** (Class membership rule). *Under $E_a^t \wedge G_a^t$ and $\forall l \in [A]$ and at time $t$: $d_{a,\delta}^t(l) > 0 \iff l \in [A] \backslash \mathcal{C}_a$.*

Using the above lemma, we obtain the following result for the time complexity of class estimation.

**Theorem 1** (*ColME* class estimation time complexity). *For any $\delta \in (0,1)$, employing Restricted-Round-Robin query strategy, we have:*

$$\mathbb{P}(\exists t > \zeta_a : \mathcal{C}_a^t \ne \mathcal{C}_a) \le \frac{\delta}{8}, \quad \text{with} \quad \zeta_a = n_{a,a}^\star + A - 1 - \sum_{l \in [A] \backslash \mathcal{C}_a} \mathbb{1}_{\{n_{a,a}^\star > n_{a,l}^\star + A - 1\}}. \tag{7}$$

The dominating term in the class estimation time complexity $\zeta_a$ for agent $a$ is equal to the number $n_{a,a}^\star$ of samples required to distinguish agent $a$ from the one who has smallest nonzero gap $\Delta_a$ to $a$, which is of order $\widetilde{O}(1/\Delta_a^2)$.[2] There is then an additional term of $A - 1$ since all others agents that are not in $\mathcal{C}_a$ could require the same number of samples. Finally, the last term in (7) accounts for agents that require less samples and had thus been eliminated before, which reflects the gain of using *Restricted-Round-Robin* query strategy over *Round-Robin*. When we have enough samples (at least $\zeta_a$), Theorem 1 guarantees that we correctly learn the class ($\mathcal{C}_a = \mathcal{C}_a^t$) with high probability. We build upon this result to quantify the mean estimation time complexity of our approach.

**Theorem 2** (*ColME* mean estimation time complexity). *Given the risk parameter $\delta$, using the Restricted-Round-Robin query strategy and simple weighting, the mean estimator $\mu_a^t$ of agent $a$ is $(\epsilon, \frac{\delta}{4})$-convergent, that is:*

$$\mathbb{P}(\forall t > \tau_a : |\mu_a^t - \mu_a| \le \epsilon) > 1 - \frac{\delta}{4}, \quad \text{with} \quad \tau_a = \max(\zeta_a, \frac{\lceil \beta_\delta^{-1}(\epsilon) \rceil}{|\mathcal{C}_a|} + \frac{|\mathcal{C}_a| - 1}{2}). \tag{8}$$

---

[2]We use $\widetilde{O}(\cdot)$ to hide constant and logarithmic terms.

Several comments are in order. First, recall that collaboration induces a bias in mean estimation before class estimation time. Because the problem structure is unknown, any collaborative algorithm that aggregate observations from different agents will suffer from this bias, but the bias vanishes as soon as the class is estimated and we outperform local estimation.

Then, to interpret the guarantees provided by Theorem 2, it is useful to compare them with the local estimation baseline, which has time complexity $\lceil \beta_\delta^{-1}(\epsilon) \rceil = \widetilde{O}(1/\epsilon^2)$. Inspecting (8), we see that our approach has a time complexity of $\widetilde{O}(\max\{1/\Delta_a^2, 1/\epsilon^2 |\mathcal{C}_a|\})$. In other words, it is faster than local estimation as long as the time $\zeta_a$ needed to correctly identify the class $\mathcal{C}_a$ is smaller than $\lceil \beta_\delta^{-1}(\epsilon) \rceil$, that is precisely when:

$$\epsilon < \beta_\delta \big( n_{a,a}^\star + A - 1 - \sum_{l \in [A] \setminus \{\mathcal{C}_a\}} \mathbb{1}_{\{n_{a,a}^\star > n_{a,l}^\star + A - 1\}} \big). \tag{9}$$

This condition, which roughly amounts to $\epsilon < \Delta_a$, relates the desired precision of the solution $\epsilon$ to the problem structure captured by the gaps $\{\Delta_{a,l}\}_{l \in [A]}$ between the true means through $\{n_{a,l}^\star\}_{l \in [A]}$ (see Definition 6). We will see in our experiments that our theory predicts quite well whether an agent empirically benefits from collaboration.

Remarkably, our approach can be up to $|\mathcal{C}_a|$ times faster than local estimation: this happens roughly when $\epsilon < \Delta_a / \sqrt{|\mathcal{C}_a|}$, i.e., for large enough gaps or small enough $\epsilon$. In this regime, *the speed-up achieved by our approach is nearly optimal.* Indeed, the time complexity of the oracle weighting baseline introduced in Section 4.3 is precisely $\frac{\lceil \beta_\delta^{-1}(\epsilon) \rceil}{|\mathcal{C}_a|} + \frac{|\mathcal{C}_a| - 1}{2} = \widetilde{O}(1/\epsilon^2 |\mathcal{C}_a|)$, just like our approach. Note that in a full information setting where agent $a$ would know $\mathcal{C}_a$ *and* would also have access to up-to-date samples from all agents at each step, the time complexity would be only slightly smaller, namely $\frac{\lceil \beta_\delta^{-1}(\epsilon) \rceil}{|\mathcal{C}_a|}$.

**Remark 2** (Frequency of communication). *For simplicity, we consider that agents communicate each time they collect a new sample, which is standard in the literature of collaborative learning (see for instance the collaborative MAB approaches discussed in Section 2). However, different trade-offs between communication and data collection can be considered. In particular, it is straightforward to adapt the setting and our results to the case where each agent collects $m$ samples between each communication: it amounts to multiplying by $m$ the number of observations in our confidence intervals and empirical estimates. This provides a way to reduce communication, as well as to mitigate privacy concerns by ensuring that only sufficiently aggregated quantities are exchanged (even in early rounds). Extensions to cases where the number of samples between each communication is random and/or varies across agents are left for future work.*

## 6 Numerical Results

In this section, we provide numerical experiments on synthetic data to illustrate our theoretical results and assess the practical performance of our proposed algorithms.[3]

### 6.1 Experimental Setting

We consider $A = 200$ agents, a time horizon of 2500 steps and a risk parameter $\delta = 0.001$. The personal distributions of agents are all Gaussian with variance $\sigma^2 = 0.25$ and belong to one of 3 classes with means 0.2, 0.4 and 0.8. The class membership of each agent (and thus the value of its true mean) is chosen uniformly at random among the three classes. We thus obtain roughly balanced class sizes. While the evaluation presented in this section focuses on this 3-class problem, in the appendix we provide additional results on a simpler 2-class problem (with means 0.2 and 0.8) where the benefits of our algorithm is even more significant.

We consider several variants of our algorithm *ColME*: we compare query strategies *Round-Robin* and *Restricted-Round-Robin* with simple weighting, and also evaluate the use of soft and aggressive weighting schemes in the *Restricted-Round-Robin* case. This gives 4 variants of our algorithm: *Round-Robin*, *Restricted-Round-Robin*, *Soft-Restricted-Round-Robin* and *Aggressive-Restricted-Round-Robin*.

Regarding competing approaches, we recall that our setting is novel and we are not aware of existing algorithms addressing the same problem. We can however compare against two baseline strategies. The *Local*

---

[3]The code can be found at https://github.com/llvllahsa/CollaborativePersonalizedMeanEstimation

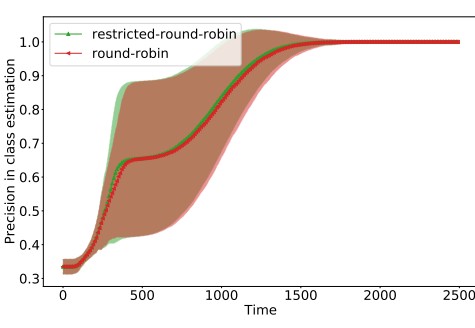

(a) Class estimation precision over all agents

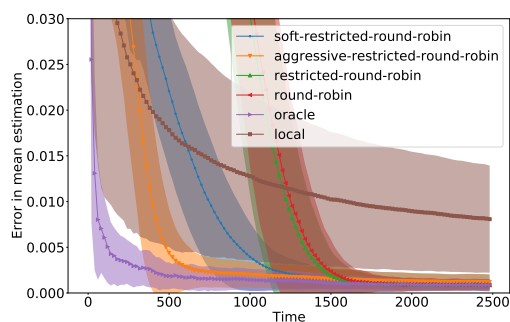

(b) Error in mean estimation over all agents

Figure 1: Results across time on the 3-class problem (Gaussian distributions with true means 0.2, 0.4, 0.8). Thanks to our collaborative algorithms (*Soft-Restricted-Round-Robin*, *Aggressive-Restricted-Round-Robin*, *Restricted-Round-Robin*, *Round-Robin*), agents are able to estimate their true class (Fig. 1(a)) and thereby obtain accurate mean estimates much more quickly than using purely local estimation (Fig. 1(b)).

baseline corresponds to the case of no collaboration. On the other hand, the *Oracle* baseline represents an upper bound on the achievable performance by any collaborative algorithm as it is given as input the true class membership of each agent and thus does not need to perform class estimation.

All algorithms are compared across 20 random runs corresponding to 20 different samples. In a given run, at each time step, each agent receives the same sample for all algorithms.

## 6.2 Class Estimation

We start by investigating the performance in class estimation. In this experiment, only *Round-Robin* and *Restricted-Round-Robin* are shown since the different weighting schemes have no effect on class estimation.

We first look at how well an agent $a$ estimates its true class $\mathcal{C}_a$ with its heuristic class $\mathcal{C}_a^t$ across time. To measure this, we consider the precision at time $t$ computed as follows:

$$\text{precision}_{\mathcal{C}_a^t} = \frac{|\mathcal{C}_a^t \cap \mathcal{C}_a|}{|\mathcal{C}_a^t|}. \tag{10}$$

We compute the average and standard deviation of (10) across runs, and then average these over all agents. Figure 1(a) shows how the precision of class estimation varies across time as agents progressively remove others from their heuristic class and eventually identify their true class. As can be seen clearly in Figure 3 in the appendix, the classes 0.2 and 0.8 are separated very early, quickly followed by 0.4 and 0.8 and finally, after sufficiently many samples have been collected, the pair with the smallest gap (0.4 and 0.2).

We also observe that *Round-Robin* and *Restricted-Round-Robin* only differ slightly in the last time steps before classes are identified. This is in line with Theorem 1, which shows that class estimation time mainly depends on $n_{a,a}^\star$, the time needed to eliminate the agent with smallest nonzero gap. This dominant term and the fact that querying an agent at time $t$ yields the full statistics of observations of this agent up to time $t$ explain that the gain of *Restricted-Round-Robin* is small compared to vanilla *Round-Robin*.

Table 1 shows statistics about the average, standard deviation and maximum time taken by an agent to correctly identify its class. As expected, agents from class 0.8 identify their class much faster as the gaps with respect to other classes are larger. Table 1 also reports the high-probability class estimation time $\zeta_a$ of *Restricted-Round-Robin* given by our theoretical analysis (Theorem 1). This theoretical value is rather close to the maximum value we observe: although these values are not directly comparable, it suggests that our analysis captures the correct order of magnitude.

| Algorithm | class 0.2 | | class 0.4 | | class 0.8 | |
|---|---|---|---|---|---|---|
| | avg/std | max | avg/std | max | avg/std | max |
| Round-Robin (RR) | $1378 \pm 194$ | 2150 | $1382 \pm 191$ | 2159 | $407 \pm 40$ | 662 |
| Restricted-RR | $1376 \pm 211$ | 2163 | $1379 \pm 210$ | 2245 | $373 \pm 55$ | 934 |
| Theoretical Restricted-RR ($\zeta_a$) | 3878 | | 3878 | | 1085 | |

Table 1: Empirical class estimation times of Round-Robin and Restricted-RR on the 3-class problem (Gaussian distributions with true means 0.2, 0.4, 0.8). We report the average, standard deviation and maximum across agents and runs. We also report the high-probability class estimation time $\zeta_a$ given by our theory.

### 6.3 Mean Estimation

We now turn to our main objective: mean estimation. The error of an agent $a$ at time $t$ is evaluated as the absolute difference of its mean estimate with its true mean:

$$\text{error}_a^t = |\mu_a^t - \mu_a|. \tag{11}$$

Similar to above, we compute the average and standard deviation of this quantity across runs, and then for each time step we report in Figure 1(b) the average of these quantities across all agents for the different algorithms (*Soft-Restricted-Round-Robin*, *Aggressive-Restricted-Round-Robin*, *Restricted-Round-Robin*, *Round-Robin*, *Oracle*, and *Local*).

As expected, all variants of *ColME* suffer from mean estimation bias in the early steps (due to optimistic class estimation). However, as the estimated class of each agent gets more precise (see Figure 1(a)), agents progressively eliminate this bias and eventually learn estimates with similar error and variance as the *Oracle* baseline. On the other hand, *Local* does not have estimation bias (hence achieves smaller error on average in early rounds) but exhibits much higher variance, and its average error converges very slowly towards zero. These results show the ability of our collaborative algorithms to construct highly accurate mean estimates much faster than without collaboration. We can also see that *Soft-Restricted-Round-Robin* and *Aggressive-Restricted-Round-Robin* converge much quicker to low error estimates than *Restricted-Round-Robin*. This shows that our proposed heuristic weighting schemes successfully reduce the relative weight given to agents that actually belong to different classes well before they are identified as such with sufficient confidence. The aggressive weighting scheme is observed to perform best in practice.

In the appendix, we plot the error in mean estimation for each class separately and observe that agents with mean 0.8 (i.e., in the class that is easiest to discriminate from others) are the fastest to reach highly accurate estimates, followed by those with mean 0.4, and finally those with mean 0.2.

Finally, we quantitatively compare the convergence time of different algorithms with an empirical measure inspired by our theoretical PAC criterion (Definition 1). We define the *empirical convergence* time of an agent as the earliest time step where the estimation error of the agent always stays lower than some $\epsilon$:

$$\text{conv}_a(\epsilon) = \min\{\tau \in \mathbb{N} : \forall t \geq \tau, \text{error}_a^t \leq \epsilon\}. \tag{12}$$

We denote by $\text{conv}(\epsilon)$ the average of the above quantity across all runs and all agents.

Table 2 reports the average, standard deviation and maximum empirical convergence time across agents and runs for two values of $\epsilon$ (we also provide per-class tables in the appendix). These values were chosen to reflect the two different regimes suggested by our theoretical analysis. Indeed, recall that our theory gives a criterion to predict whether our collaborative algorithms will outperform the *Local* baseline: this is the case when the desired accuracy of the solution $\epsilon$ is small enough for the given problem instance (see Eq. 9). For the problem considered here, Eq. (9) gives that *Restricted-Round-Robin* will outperform *Local* for all agents as soon as $\epsilon$ is smaller than 0.049. We thus choose $\epsilon = 0.01$ as the favorable regime (where we should beat *Local*) and $\epsilon = 0.1$ as the unfavorable regime. We provide the corresponding high-probability mean estimation times $\tau_a$ of *Restricted-Round-Robin* and *Local* given by our theoretical analysis (Theorem 2).

| Algorithm | conv(0.1) | | conv(0.01) | |
|---|---|---|---|---|
| | avg/std | max | avg/std | max |
| Round-Robin (RR) | $417 \pm 230$ | 1239 | $916 \pm 427$ | 2088 |
| Restricted-RR | $405 \pm 227$ | 1175 | $894 \pm 438$ | 1925 |
| Soft-Restricted-RR | $82 \pm 48$ | 340 | $608 \pm 290$ | 1432 |
| Aggressive-Restricted-RR | $56 \pm 39$ | 340 | $\mathbf{335 \pm 127}$ | **958** |
| Local | $\mathbf{41 \pm 39}$ | **289** | $4494 \pm 3945$ | 28616 |
| Oracle | *5 ± 4* | *19* | *98 ± 63* | *303* |
| Theoretical Restricted-RR ($\tau_a$) | 3878 | | 33406 | |
| Theoretical Local ($\tau_a$) | 885 | | 100216 | |

Table 2: Empirical convergence times (see Eq. 12) of different algorithms on the 3-class problem (Gaussian distributions with true means 0.2, 0.4, 0.8) for a target estimation error of $\epsilon = 0.1$ (unfavorable regime) and $\epsilon = 0.01$ (favorable regime). We report the average, standard deviation and maximum across agents and runs. We also report the high-probability mean estimation times $\tau_a$ given by our theory for *Restricted-Round-Robin* and *Local*. In line with our theory, we see that our approach largely outperforms the local estimation baseline in the favorable regime and remains competitive in the unfavorable regime.

The results in Table 2 are consistent with our theory. All variants of our algorithms largely outperform *Local* for $\epsilon = 0.01$,[4] while *Local* is better for $\epsilon = 0.1$ as agents can reach this precision using only their own samples faster than they can reliably estimate their class. Overall, *Restricted-Round-Robin* performs marginally better than *Round-Robin*, while *Soft-Restricted-Round-Robin* and *Aggressive-Restricted-Round-Robin* significantly outperform *Round-Robin* and *Restricted-Round-Robin* in both cases. Note that *Soft-Restricted-Round-Robin* and *Aggressive-Restricted-Round-Robin* perform roughly the same as *Local* in the unfavorable regime, and get very close to the performance of the *Oracle* baseline in the favorable regime. These results again show the relevance of our collaborative algorithms and heuristic weighting schemes. We observe that our theoretical results get looser as $\epsilon \to 0$, which is somewhat expected.

## 7 Extension to Imperfect Classes

So far we have assumed that two agents are in the same class if their personal distributions have *exactly* the same mean, which can be restrictive for some use-cases. In this section, we show that we can extend the problem setup and our approach to the case where *two agents are considered to be in the same class if their means are close enough* and agents seek to *estimate the mean of their class.*

Formally, we define a new notion of similarity class parameterized by a radius $\eta$, which generalizes our previous notion introduced in Definition 2.

**Definition 7.** *Given $\eta > 0$, the $\eta$-similarity class of agent $a$ is given by:*

$$\mathcal{C}_{\eta,a} = \{l \in [A] : \Delta_{a,l} \leq \eta\},$$

*where $\Delta_{a,l} = |\mu_a - \mu_l|$ is the gap between the means of agent $a$ and agent $l$.*

This notion of "imperfect" similarity class allows to capture situations where *clusters* of agents have similar (but not necessarily equal) means. Such discrepancies between the means of agents in the same class may for instance stem from the presence of local measurement bias (e.g., due to local variations in the environment, see Taghavi et al., 2016). They can also be used to model groups of agents with similar preferences, behavior, or goals, in applications like collaborative filtering (Su & Khoshgoftaar, 2009b),

---

[4]We had to run *Local* for a much larger time horizon of 30000 steps for all runs to converge to accuracy $\epsilon = 0.01$.

In this context, it is natural to slightly redefine the estimation objective. Instead of estimating its personal mean $\mu_a$ as considered so far, each agent $a$ aims to estimate the mean of its class:

$$\mu_{\eta,a} = \frac{1}{|\mathcal{C}_{\eta,a}|} \sum_{l \in \mathcal{C}_{\eta,a}} \mu_l. \tag{13}$$

For instance, in the presence of (centered) local measurement bias, estimating the class mean (instead of the local mean) allows to debias the estimate.

**Remark 3** (Non-separated clusters). *We do not formally require that the $\eta$-similarity classes form separated clusters, in the sense that for three distinct agents $a, l, i \in [A]$ we may have simultaneously $i \in \mathcal{C}_{\eta,a}$, $i \in \mathcal{C}_{\eta,l}$ and $\mathcal{C}_{\eta,a} \neq \mathcal{C}_{\eta,l}$. This happens when $\Delta_{a,i} \leq \eta$, $\Delta_{l,i} \leq \eta$ and $\eta < \Delta_{a,l} \leq 2\eta$. In this case, the "class" of an agent simply corresponds to a ball of radius $\eta$ around its mean, which potentially overlaps with others and thus violates the transitivity property of equivalence classes. For consistency with the rest of the paper and with an slight abuse of terminology, we continue to use the term "class". Although the case of separated clusters appears more natural, we note that our proposed approach will still work in the non-separated setting, in the sense that agents will correctly estimate the mean of their class as defined in Eq. 13.*

Based on the above, we can adapt the notion of optimistic similarity class (Definition 2) and the condition on the number of samples required for this optimistic class to coincide with the true class (Definition 6) by incorporating $\eta$.

**Definition 8.** *The $\eta$-optimistic similarity class from the perspective of agent $a$ at time $t$ is defined as:*

$$\mathcal{C}_{\eta,a}^t = \{l \in [A] : d_{a,\delta}^t(l) \leq \eta\}.$$

**Definition 9.** *From the perspective of agent $a$ and at time $t$, event $G_{\eta,a}^t$ is defined as:*

$$G_{\eta,a}^t = \bigcap_{l \in [A]} n_{a,l}^t > n_{a,l}^\eta, \tag{14}$$

$$where \qquad n_{a,l}^\eta = \begin{cases} \lceil \beta_\delta^{-1}(\frac{\Delta_{a,l}-\eta}{4}) \rceil & if\ l \notin \mathcal{C}_a, \\ \lceil \beta_\delta^{-1}(\frac{\Delta_{\eta,a}-\eta}{4}) \rceil & otherwise, \end{cases}$$

*with $\Delta_{\eta,a} = \min_{l \in [A] \setminus \mathcal{C}_{\eta,a}} \Delta_{a,l}$.*

**Lemma 3** (Class membership rule). *Under $E_a^t \wedge G_{\eta,a}^t$ and $\forall l \in [A]$ and at time $t$: $d_{a,\delta}^t(l) > \eta \iff l \in [A] \setminus \mathcal{C}_{\eta,a}$.*

We can see from the above that ruling out an agent $l$ from the optimistic class $\mathcal{C}_{\eta,a}$ requires more samples for larger $\eta$, which is expected as the size of the confidence interval needs to be smaller to make this decision reliably.

With these tools in place, we can use our collaborative mean estimation algorithm ColME (Algorithm 1) presented before, with only minor modifications: we simply need to replace the notion of optimistic similarity class by the $\eta$-version of Definition 8, and compute the estimate $\mu_{\eta,a}^t$ at time $t$ using a simple *class-uniform weighting scheme* $\alpha_{a,l}^t = \frac{1}{|\mathcal{C}_{\eta,a}^t|}$ to match the objective in Eq. 13. We refer to this algorithm as *$\eta$-ColME*. Note that $\eta$ becomes a parameter of the algorithm, allowing to choose the desired radius for the class structure.

We can now state the class and mean estimation complexity of *$\eta$-ColME*. The proofs can be found in the appendix.

**Theorem 3** (*$\eta$-ColME* class estimation time complexity). *For any $\delta \in (0,1)$, employing Restricted-Round-Robin query strategy, we have:*

$$\mathbb{P}\left(\exists t > \zeta_a^\eta : \mathcal{C}_{\eta,a}^t \neq \mathcal{C}_{\eta,a}\right) \leq \frac{\delta}{8}, \quad with \quad \zeta_a^\eta = n_{a,a}^\eta + A - 1 - \sum_{l \in [A] \setminus \mathcal{C}_{\eta,a}} \mathbb{1}_{\{n_{a,a}^\eta > n_{a,l}^\eta + A - 1\}}. \tag{15}$$

**Theorem 4** ($\eta$-*ColME* mean estimation time complexity). *Given the risk parameter $\delta$, using the Restricted-Round-Robin query strategy and class-uniform weighting (while employing $\mathcal{C}_{\eta,a}$), the mean estimator $\mu_a^t$ of agent $a$ is $(\epsilon, \frac{\delta}{4})$-convergent, that is:*

$$\mathbb{P}\big(\forall t > \tau_a^\eta : |\mu_{\eta,a}^t - \mu_{\eta,a}| \le \epsilon\big) > 1 - \frac{\delta}{4}, \quad with \quad \tau_a^\eta = \max(\zeta_a^\eta, \beta_\delta^{-1}(\epsilon) + |\mathcal{C}_{\eta,a}| - 1). \tag{16}$$

The class estimation result (Theorem 3) is similar to its "perfect" class counterpart (Theorem 1) except that it involves $\eta$-dependent quantities. The mean estimation result (Theorem 4) differs slightly more from the perfect class case (Theorem 2) because the estimation objective (see Eq. 13) and weighting scheme are different. But most importantly, we see that for large enough gaps or small enough precision $\epsilon$ (similar to Eq. 9), we again achieve an optimal speed since the time complexity of an oracle weighting baseline that would know the true classes beforehand is $\beta_\delta^{-1}(\epsilon) + |\mathcal{C}_{\eta,a}| - 1$.

## 8 Conclusion

We have presented collaborative online algorithms where each agent learns the set (class) of agents who shares the same (or similar) objective and uses this knowledge to speed up the estimation of its personalized mean. We have provided PAC-style guarantees for the class and mean estimation time complexity of our algorithms. In addition, we have introduced a number of sample weighting mechanisms to decrease the bias of the estimates in the early rounds of learning, whose benefits are illustrated empirically.

Our work initiates the study of online, collaborative and personalized estimation and learning problems, which we believe to be a promising area for future work. We outline a few interesting directions below.

**Optimistic vs conservative.** Instead of the optimistic approach taken in this work, one could try to design a more conservative algorithm where an agent incorporates the estimate of another agent only when it knows (with sufficient probability) that it belongs to the same class. This would avoid introducing bias in the estimation in early steps. However, a downside of such an approach is that agents would typically need some knowledge of the gaps between their true means in order to determine when another agent can be considered to be in the same class, which would be a big practical limitation.

**Large-scale variants.** While a simple way to scale up our approach to a large number of agents is to have each agent focus on a restricted subset of other agents (see Remark 1), an interesting direction to allow more exploration in large-scale systems could rely on the idea of peer sampling (Jelasity et al., 2007), i.e., randomly sampling a few agents from time to time to discover potential new members of the class beyond the initial subset.

**Handling data drift.** We would like to extend our approach to handle data drift, where the means of agents can change over time. Here, one could try to adapt ideas from non-stationary bandits, such as sliding-window UCB (Garivier & Moulines, 2011) or UCB strategies mixed with change-point detection algorithms (Cao et al., 2019).

**Privacy guarantees.** In use cases where data is considered sensitive (e.g., personal data), it is important to provide strong privacy guarantees to the agents. While our approach only requires agents to share aggregate quantities (see also Remark 2), these may still leak sensitive information. In future work, we would like to use tools from differential privacy (Dwork & Roth, 2013), such as the tree aggregation technique for sharing cumulative sums in an online way (Dwork et al., 2010; Chan et al., 2011), to provide formal privacy guarantees and analyze the resulting trade-offs between privacy and utility.

**Extensions to personalized learning tasks.** Finally, the problem could be extended to cases where each agent aims to solve a personalized machine learning task (Vanhaesebrouck et al., 2017) based on the data it receives online. In that case, a structure in the distribution conditioned by the outputs of the learned models can be inferred, introducing an interesting exploration-exploitation dilemma in the learning task.

**Acknowledgments**

The authors thank the reviewers for their insightful comments that allowed to improve the paper. This work was funded in part by Métropole Européenne de Lille (MEL), Inria, Université de Lille, and the I-SITE ULNE through the AI chair Apprenf number R-PILOTE-19-004-APPRENF.

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

## Appendix A    Proof of Lemma 1

**Lemma 4.** *Let $\mu_a^t$ be the mean value of $t$ independent real-valued random variables with the true mean $\mu_a$ and is $\sigma$-sub-gaussian. For all $\delta \in (0,1)$, it holds:*

$$\mathbb{P}\left(\exists t \in \mathbb{N}, \mu_a^t - \mu_a \geq \sigma\sqrt{\frac{2}{t}(1+\frac{1}{t})\ln(\sqrt{t+1}/\delta)}\right) \leq \delta \ , \tag{17}$$

$$\mathbb{P}\left(\exists t \in \mathbb{N}, \mu_a - \mu_a^t \geq \sigma\sqrt{\frac{2}{t}(1+\frac{1}{t})\ln(\sqrt{t+1}/\delta)}\right) \leq \delta \ . \tag{18}$$

*Proof.* The two inequalities are proved in the same way as a direct consequence of Maillard (2019), Lemma 2.7 therein. Let $Y_1, \ldots, Y_t$ be a sequence of independent real-valued random variables where for each $s \leq t$, $Y_s$ has mean $\mu_s$ and is $\sigma_s$-sub-Gaussian, then for all $\delta \in (0,1)$, it holds that

$$\mathbb{P}\left(\exists t \in \mathbb{N}, \sum_{s=1}^{t}(Y_s - \mu_s) \geq \sqrt{2\sum_{s=1}^{t}\sigma_s^2(1+\frac{1}{t})\ln(\sqrt{t+1}/\delta)}\right) \leq \delta \ .$$

When all random variables $Y_s$ have the same mean $\mu_a$ and variance $\sigma$, we have

$$\mathbb{P}\left(\exists t \in \mathbb{N}, \sum_{s=1}^{t}(Y_s - \mu_a) \geq \sqrt{2t\sigma^2(1+\frac{1}{t})\ln(\sqrt{t+1}/\delta)}\right) \leq \delta,$$

Taking the average rather than the sum, i.e. dividing both sides by $t$ we obtain that:

$$\mathbb{P}\left(\exists t \in \mathbb{N}, \sum_{s=1}^{t}(\frac{Y_s}{t} - \frac{\mu_a}{t}) \geq \sqrt{\frac{2}{t}\sigma^2(1+\frac{1}{t})\ln(\sqrt{t+1}/\delta)}\right) \leq \delta,$$

$$\mathbb{P}\left(\exists t \in \mathbb{N}, \sum_{s=1}^{t}\frac{Y_s}{t} - \mu_a \geq \sqrt{\frac{2}{t}\sigma^2(1+\frac{1}{t})\ln(\sqrt{t+1}/\delta)}\right) \leq \delta \ .$$

And denoting $\mu_a^t = \sum_{s=1}^{t}\frac{Y_s}{t}$, we conclude

$$\mathbb{P}\left(\exists t \in \mathbb{N}, \mu_a^t - \mu_a \geq \sigma\sqrt{\frac{2}{t}(1+\frac{1}{t})\ln(\sqrt{t+1}/\delta)}\right) \leq \delta \ .$$

$\square$

**Lemma 1.** *Let $\delta \in (0,1)$, $a \in [A]$. Setting $\beta_\delta(n) = \sigma\sqrt{2\frac{1}{n} \times (1+\frac{1}{n})\ln(\sqrt{n+1}/\gamma(\delta))}$ with $\gamma(\delta) = \frac{\delta}{8 \times A}$, we have:*

$$\mathbb{P}(E_a) \geq 1 - \frac{\delta}{8}. \tag{5}$$

*Proof.* Let us recall that $E_a = \bigcap_{t \in \mathbb{N}} \bigcap_{l \in [A]} |\bar{x}_{a,l}^t - \mu_l| \leq \beta_\delta(n_{a,l}^t)$. Then

$$\begin{aligned}
\mathbb{P}(E_a) &= 1 - \mathbb{P}(\bar{E}_a), \\
&= 1 - \mathbb{P}\left(\exists t \in \mathbb{N}, \exists l \in [A] : |\bar{x}_{a,l}^t - \mu_l| > \beta_\delta(n_{a,l}^t)\right), \\
&\geq 1 - \sum_{l \in [A]} \mathbb{P}\left(\exists t \in \mathbb{N} : |\bar{x}_{a,l}^t - \mu_l| > \beta_\delta(n_{a,l}^t)\right).
\end{aligned}$$

defining $\gamma(\delta) = \frac{\delta}{8 \times A}$ and using Lemma 4

$$\mathbb{P}(E_a) \geq 1 - \sum_{l \in [A]} \mathbb{P}\left(\exists t \in \mathbb{N} : |\bar{x}_{a,l}^t - \mu_l| > \sigma\sqrt{\frac{2}{n_{a,l}^t} \times (1+\frac{1}{n_{a,l}^t})\ln(\sqrt{n_{a,l}^t+1}/\gamma(\delta))}\right),$$

$$\geq 1 - \sum_{l \in [A]} \gamma(\delta) = 1 - \sum_{l \in [A]} \frac{\delta}{8A} = 1 - \frac{\delta}{8}. \qquad \square$$

# Appendix B    Proof of Theorem 1

In this section, we prove Theorem 1. We first show Lemma 2 using two technical lemmas. We then prove Lemma 7, which combined with Lemma 2, yields the main result (Theorem 1). Let us first remark that

$$d_{a,\delta}^t(l) = |\bar{x}_{a,a}^t - \bar{x}_{a,l}^t| - \beta_\delta(n_{a,a}^t) - \beta_\delta(n_{a,l}^t),$$
$$= |(\bar{x}_{a,a}^t - \mu_a) - (\bar{x}_{a,l}^t - \mu_l) + (\mu_a - \mu_l)| - \beta_\delta(n_{a,a}^t) - \beta_\delta(n_{a,l}^t).$$

As a consequence we have

$$d_{a,\delta}^t(l) \le \Delta_{a,l} + |\bar{x}_{a,a}^t - \mu_a| + |\bar{x}_{a,l}^t - \mu_l| - \beta_\delta(n_{a,a}^t) - \beta_\delta(n_{a,l}^t). \tag{19}$$
$$d_{a,\delta}^t(l) \ge \Delta_{a,l} - |\bar{x}_{a,a}^t - \mu_a| - |\bar{x}_{a,l}^t - \mu_l| - \beta_\delta(n_{a,a}^t) - \beta_\delta(n_{a,l}^t). \tag{20}$$

**Lemma 5.** *Under $E_a$, $\forall l \in [A]$, if $l \notin \mathcal{C}_a$ then $\forall n_{a,l}^t \ge n_{a,l}^\star = \lceil \beta_\delta^{-1}(\frac{\Delta_{a,l}}{4}) \rceil$ we have $d_{a,\delta}^t(l) > 0$.*

*Proof.* Under $E_a$, we have $|\bar{x}_{a,l}^t - \mu_l| \le \beta_\delta(n_{a,l}^t)$ and $|\bar{x}_{a,a}^t - \mu_a| \le \beta_\delta(n_{a,a}^t)$. Since $n_{a,a}^t \ge n_{a,l}^t$, we also have $\beta_\delta(n_{a,a}^t) \le \beta_\delta(n_{a,l}^t)$. Hence, using (20), $d_{a,\delta}^t(l) \ge \Delta_{a,l} - 2\beta_\delta(n_{a,a}^t) - 2\beta_\delta(n_{a,l}^t) \ge \Delta_{a,l} - 4\beta_\delta(n_{a,l}^t)$. If $l \in \mathcal{C}_a$ then $\Delta_{a,l} = 0$ and since $\beta_\delta(n_{a,l}^t) > 0$ we cannot ensure that $\Delta_{a,l} - 4\beta_\delta(n_{a,l}^t) > 0$. If $l \notin \mathcal{C}_a$ then to ensure that $d_{a,\delta}^t(l) \ge \Delta_{a,l} - 4\beta_\delta(n_{a,l}^t) > 0$, we need that $\frac{\Delta_{a,l}}{4} > \beta_\delta(n_{a,l}^t)$ and hence $n_{a,l}^\star = \lceil \beta_\delta^{-1}(\frac{\Delta_{a,l}}{4}) \rceil$.[5]    □

**Lemma 6.** *Under $E_a$, $\forall l \in [A]$, $\forall t \in \mathbb{N}$, if $l \in \mathcal{C}_a$ then $d_{a,\delta}^t(l) \le 0$.*

*Proof.* Again, recall that under $E_a^t$, we have $|\bar{x}_{a,l}^t - \mu_l| \le \beta_\delta(n_{a,l}^t)$ and $|\bar{x}_{a,a}^t - \mu_a| \le \beta_\delta(n_{a,a}^t)$. Hence, using (19), $d_{a,\delta}^t(l) \le \Delta_{a,l} + \beta_\delta(n_{a,a}^t) + \beta_\delta(n_{a,l}^t) - \beta_\delta(n_{a,a}^t) - \beta_\delta(n_{a,l}^t) = \Delta_{a,l}$. If $l \in \mathcal{C}_a$ then $\Delta_{a,l} = 0$ and thus $d_{a,\delta}^t(l) \le 0$.

□

We can now prove Lemma 2, which we restate here for convenience.

**Lemma 2** (Class membership rule). *Under $E_a^t \wedge G_a^t$ and $\forall l \in [A]$ and at time $t$: $d_{a,\delta}^t(l) > 0 \iff l \in [A] \backslash \mathcal{C}_a$.*

*Proof.* From Lemma 6, we directly have one implication. For the other one, if $l \notin \mathcal{C}_a$ because $G_a^t$ holds, we have $\forall l \in [A]$, $n_{a,l}^t \ge n_{a,l}^\star$, therefore we can apply Lemma 5 and we directly have $d_{a,\delta}^t(l) > 0$.    □

**Lemma 7.** *Under $E_a$, and using Restricted-Round-Robin algorithm, $G_a^t$ holds when $t > \zeta_a$ where*

$$\zeta_a = n_{a,a}^\star - 1 + A - \sum_{l \in [A] \backslash \mathcal{C}_a} \mathbb{1}_{\{n_{a,a}^\star > n_{a,l}^\star - 1 + A\}}.$$

*Proof.* According to Algorithm 1, $\mathcal{C}_a^0 = [A]$ and an agent is eliminated from set $\mathcal{C}_a^t$ at time $t$ if $d_{a,\delta}^t(l) = |\bar{x}_{a,l}^t - \bar{x}_{a,a}^t| - \beta_\delta(n_{a,a}^t) - \beta_\delta(n_{a,l}^t) > 0$. According to Lemma 5, the time required to eliminate agent $l$ from the class $\mathcal{C}_a$ is at least $n_{a,l}^\star$. If agent $l$ is queried at time $n_{a,l}^\star - 1$, then using *Restricted-Round-Robin* (or round robin), we are sure that it will be removed from $\mathcal{C}_a^t$ for all $t$ larger than $n_{a,l}^\star - 1 + A$.

Let us consider $h$ being an agent such that $n_{a,h}^\star = \max_{l \in [A] \backslash \mathcal{C}_a} n_{a,l}^\star$. By definition, $\Delta_{a,h} = \min_{l \in [A] \backslash \mathcal{C}_a} \Delta_{a,l}$ and $n_{a,h}^\star$ can be denoted by $n_{a,a}^\star$.

In the case of round robin, we are sure that $G_a^t$ will be true when $t \ge n_{a,a}^\star - 1 + A$. But using *Restricted-Round-Robin*, since the loop ignores agents not in $\mathcal{C}_a^t$, we have that $G_a^t$ holds when $t > \zeta_a$ where

---

[5]In extremely rare cases, the expression $\beta_\delta^{-1}(\frac{\Delta_{a,l}}{4})$ could be an integer and we should add 1 to get a strict inequality. But for conciseness of the expression, we omit the $+1$ in the definition of $n_{a,l}^\star$.

$$\zeta_a = n_{a,a}^{\star} - 1 + A - \sum_{l \in [A] \setminus \mathcal{C}_a} \mathbb{1}_{\{n_{a,a}^{\star} > n_{a,l}^{\star} - 1 + A\}}. \hspace{2cm} \square$$

Finally, we use the above lemmas to prove Theorem 1, which we restate below for convenience.

**Theorem 1** (*ColME* class estimation time complexity). *For any $\delta \in (0,1)$, employing Restricted-Round-Robin query strategy, we have:*

$$\mathbb{P}\left(\exists t > \zeta_a : \mathcal{C}_a^t \neq \mathcal{C}_a\right) \leq \frac{\delta}{8}, \quad \text{with} \quad \zeta_a = n_{a,a}^{\star} + A - 1 - \sum_{l \in [A] \setminus \mathcal{C}_a} \mathbb{1}_{\{n_{a,a}^{\star} > n_{a,l}^{\star} + A - 1\}}. \tag{7}$$

*Proof.* From Lemma 7 and Lemma 2, we deduce that if $E_a$ holds and knowing that $\mathcal{C}_a^t = \{l \in [A] : d_{a,\delta}^t(l) \leq 0\}$ then $\forall t > \zeta_a$, $\mathcal{C}_a = \mathcal{C}_a^t$. Hence, $\mathbb{P}\left(\forall t > \zeta_a, \mathcal{C}_a = \mathcal{C}_a^t\right) \geq \mathbb{P}\left(E_a\right) \geq 1 - \delta/8$ using Lemma 1. $\hspace{1cm}\square$

## Appendix C   Proof of Theorem 2

In this section we detail the proof of Theorem 2, about the PAC mean estimation properties of the ColME strategy. We restate the theorem below for convenience.

**Theorem 2** (*ColME* mean estimation time complexity). *Given the risk parameter $\delta$, using the Restricted-Round-Robin query strategy and simple weighting, the mean estimator $\mu_a^t$ of agent $a$ is $(\epsilon, \frac{\delta}{4})$-convergent, that is:*

$$\mathbb{P}\left(\forall t > \tau_a : |\mu_a^t - \mu_a| \leq \epsilon\right) > 1 - \frac{\delta}{4}, \quad \text{with} \quad \tau_a = \max(\zeta_a, \frac{\lceil \beta_\delta^{-1}(\epsilon) \rceil}{|\mathcal{C}_a|} + \frac{|\mathcal{C}_a| - 1}{2}). \tag{8}$$

*Proof.* Let us assume that at time $t$ we have $\mathcal{C}_a^t = \mathcal{C}_a$. Therefore

$$\mu_a^t = \sum_{l \in \mathcal{C}_a} \bar{x}_{a,l}^t \alpha_{a,l}^t = \frac{\sum_{l \in \mathcal{C}_a} \bar{x}_{a,l}^t n_{a,l}^t}{\sum_{l \in \mathcal{C}_a} n_{a,l}^t}.$$

Remark that $\sum_{l \in \mathcal{C}_a} \bar{x}_{a,l}^t n_{a,l}^t$ is the sum of all $n_{a,l}^t$ samples received by all agents $l$ in $\mathcal{C}_a$. In other words, $\mu_a^t$ is the estimation of $\mu_a$ with $\sum_{l \in \mathcal{C}_a} n_{a,l}^t$ examples. Hence in order to have $|\mu_a^t - \mu_a| \leq \epsilon$ when $E_a$ holds, we should have $\beta(\sum_{l \in \mathcal{C}_a} n_{a,l}^t) \leq \epsilon$. Let us see at what time denoted by $n_{\epsilon,a}$ we have $\lceil \beta^{-1}(\epsilon) \rceil = \sum_{l \in \mathcal{C}_a} n_{a,l}^t$. With Algorithm 13 using *Restricted-Round-Robin*, we know that when $\mathcal{C}_a^t = \mathcal{C}_a$, then only members of $\mathcal{C}_a$ are queried. Therefore,

$$\lceil \beta^{-1}(\epsilon) \rceil = n_{\epsilon,a} + (n_{\epsilon,a} - 1) + \cdots + (n_{\epsilon,a} - |\mathcal{C}_a| + 1) = |\mathcal{C}_a| n_{\epsilon,a} - \frac{|\mathcal{C}_a| - 1}{2} |\mathcal{C}_a|,$$

$$n_{\epsilon,a} = \frac{\lceil \beta^{-1}(\epsilon) \rceil}{|\mathcal{C}_a|} + \frac{|\mathcal{C}_a| - 1}{2}.$$

As a summary, if $E_a$ holds, then we have $\forall t \geq n_{\epsilon,a}$, $\mathcal{C}_a^t = \mathcal{C}_a$ implies that $|\mu_a^t - \mu_a| \leq \epsilon$. Now, following Theorem 1, we have $\mathbb{P}\left(\exists t > \zeta_a : \mathcal{C}_a^t \neq \mathcal{C}_a\right) \leq \frac{\delta}{8}$. Since $\tau_a = \max(\zeta_a, n_{\epsilon,a}) \geq \zeta_a$, then $\mathbb{P}\left(\exists t > \tau_a : |\mu_a^t - \mu_a| > \epsilon\right) \leq \frac{\delta}{8} + \mathbb{P}\left(\bar{E}_a\right) = \frac{\delta}{4}$. $\hspace{0.5cm}\square$

## Appendix D   Proof of Theorem 3

The proof of Theorem 3 follows the same step as that of Theorem 1, up to replacing the 0 threshold by $\eta$. We only state the intermediate lemmas (which are adaptations of Lemmas 5-6-7) and omit the detailed proof.

**Lemma 8.** *Under $E_a$, $\forall l \in [A]$, if $l \notin \mathcal{C}_{\eta,a}$ then $\forall n_{a,l}^t \geq n_{a,l}^\eta = \lceil \beta_\delta^{-1}(\frac{\Delta_{a,l} - \eta}{4}) \rceil$ we have $d_{a,\delta}^t(l) > \eta$.*

**Lemma 9.** *Under $E_a$, $\forall l \in [A]$, $\forall t \in \mathbb{N}$, if $l \in \mathcal{C}_{\eta,a}$ then $d_{a,\delta}^t(l) \le \eta$.*

**Lemma 10.** *Under $E_a$, and using Restricted-Round-Robin algorithm, $G_{\eta,a}^t$ holds when $t > \zeta_a^\eta$ where*

$$\zeta_a^\eta = n_{a,a}^\eta - 1 + A - \sum_{l \in [A] \setminus \mathcal{C}_{\eta,a}} \mathbb{1}_{\{n_{a,a}^\eta > n_{a,l}^\eta - 1 + A\}}.$$

## Appendix E    Proof of Theorem 4

In this section we detail the proof of Theorem 4, about the PAC mean estimation properties of the $\eta$-ColME strategy. We restate the theorem below for convenience.

**Theorem 4** ($\eta$-*ColME* mean estimation time complexity). *Given the risk parameter $\delta$, using the Restricted-Round-Robin query strategy and class-uniform weighting (while employing $\mathcal{C}_{\eta,a}$), the mean estimator $\mu_a^t$ of agent $a$ is $(\epsilon, \frac{\delta}{4})$-convergent, that is:*

$$\mathbb{P}\big(\forall t > \tau_a^\eta : |\mu_{\eta,a}^t - \mu_{\eta,a}| \le \epsilon\big) > 1 - \frac{\delta}{4}, \quad with \ \ \tau_a^\eta = \max(\zeta_a^\eta, \beta_\delta^{-1}(\epsilon) + |\mathcal{C}_{\eta,a}| - 1). \tag{16}$$

*Proof.* Since $t > \tau_a^\eta > \zeta_a^\eta$, at time $t$ we have $\mathcal{C}_{\eta,a}^t = \mathcal{C}_{\eta,a}$ . Therefore

$$\mu_a^t = \sum_{l \in \mathcal{C}_{\eta,a}} \bar{x}_{a,l}^t \alpha_{a,l}^t = \frac{\sum_{l \in \mathcal{C}_{\eta,a}} \bar{x}_{a,l}^t}{|\mathcal{C}_{\eta,a}|}.$$

Remark that $\mu_a^t$ is not equivalent to the average of all the samples of agents in $\mathcal{C}_{\eta,a}$: it is the average of the mean values for each agent in $\mathcal{C}_{\eta,a}$. Therefore, although some agents may have more samples than the others, all are assigned uniform weights. We would like to have $|\mu_{\eta,a}^t - \mu_{\eta,a}| \le \epsilon$. When $E_a$ holds, we can rewrite this as

$$|\mu_{\eta,a}^t - \mu_{\eta,a}| = \Big|\frac{1}{|\mathcal{C}_{\eta,a}|} \sum_{l \in \mathcal{C}_{\eta,a}} \bar{x}_{a,l}^t - \mu_l\Big| \le \frac{1}{|\mathcal{C}_{\eta,a}|} \sum_{l \in \mathcal{C}_{\eta,a}} |\bar{x}_{a,l}^t - \mu_l| \le \epsilon$$

Therefore, we need:

$$\sum_{l \in \mathcal{C}_{\eta,a}} |\bar{x}_{a,l}^t - \mu_l| \le |\mathcal{C}_{\eta,a}| \times \epsilon,$$

A sufficient condition for the above inequality to hold is to ensure that each term is bounded by $\epsilon$:

$$\forall l \in \mathcal{C}_{\eta,a} : |\bar{x}_{a,l}^t - \mu_l| \le \epsilon \tag{21}$$

This is achieved when $\beta_\delta(n_{a,l}^t) < \epsilon$ for all $l \in \mathcal{C}_{\eta,a}$. Since we are using *Restricted-Round-Robin* and also that $\mathcal{C}_{\eta,a}^t = \mathcal{C}_{\eta,a}$, the number of samples required for each agent in $\mathcal{C}_{\eta,a}$ are $n_{a,1}^t, n_{a,1}^t - 1, n_{a,1}^t - 2, \ldots, n_{a,1}^t - |\mathcal{C}_{\eta,a}| + 1$ where we consider the one with the maximum number of observations to have index 1 for notation simplicity (which corresponds to index $a$). For Eq. 21 to hold, it is thus sufficient to have:

$$\beta_\delta^{-1}(\epsilon) < n_{a,a}^t - |\mathcal{C}_{\eta,a}| + 1$$

$$\beta_\delta^{-1}(\epsilon) + |\mathcal{C}_{\eta,a}| - 1 < n_{a,a}^t$$

Therefore $\tau_a^\eta = \max(\zeta_a^\eta, \beta_\delta^{-1}(\epsilon) + |\mathcal{C}_{\eta,a}| - 1)$.

As a summary, if $E_a$ holds, then we have $\forall t \ge \tau_a^\eta, \mathcal{C}_{\eta,a}^t = \mathcal{C}_{\eta,a}$ implies that $|\mu_{\eta,a}^t - \mu_{\eta,a}| \le \epsilon$. Now, following Theorem 3, we have $\mathbb{P}\big(\exists t > \zeta_a^\eta : \mathcal{C}_{\eta,a}^t \ne \mathcal{C}_{\eta,a}\big) \le \frac{\delta}{8}$. Since $\tau_a^\eta = \max(\zeta_a^\eta, n_{\epsilon,a}^\eta) \ge \zeta_a^\eta$, then $\mathbb{P}\big(\exists t > \tau_a^\eta : |\mu_a^t - \mu_{\eta,a}| > \epsilon\big) \le \frac{\delta}{8} + \mathbb{P}\big(\bar{E}_a\big) = \frac{\delta}{4}$. $\square$

## Appendix F    Additional Experimental Results

In this section we provide additional illustrative results to better understand different aspects of our *ColME* algorithm.

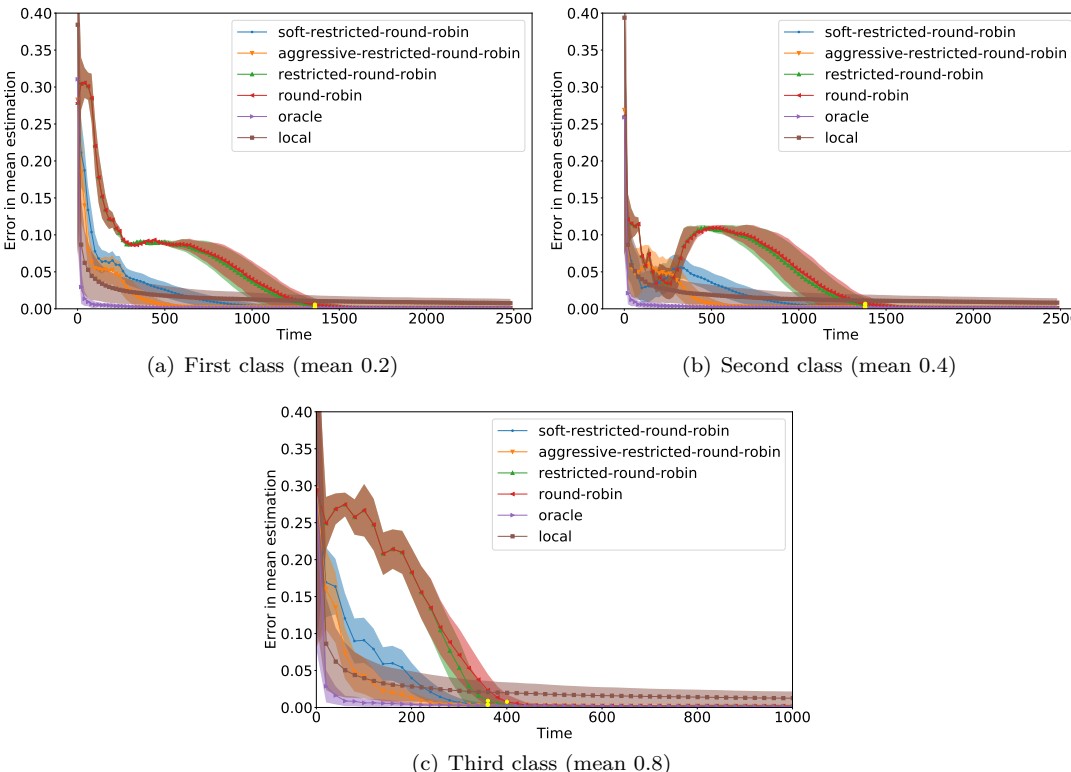

(a) First class (mean 0.2)

(b) Second class (mean 0.4)

(c) Third class (mean 0.8)

Figure 2: Error in mean estimation for the 3-class problem (Gaussian distributions with true means 0.2, 0.4, 0.8). Note that the time scale is different for the third class to show the relevant details more clearly.

### F.1 Per-Class behavior on the 3-class problem

For the 3-class problem described in the main text, we provide complementary figures to show the error in mean estimation *in each class separately*. These plots are shown in Figure 2. We can see that the average class identification time (represented by yellow dots) is different for different classes. For instance, as the gap between the class with mean 0.8 and the other two is larger, this class requires less samples to be identified. Indeed, the average class identification time is less than $t = 400$ for that class (Figure 2(c)), while it is about $t = 1400$ for the other two (Figures 2(a)-2(b)). Therefore, agents from the class with mean 0.8 reach a highly accurate estimate much faster than agents from other classes.

We also show the class estimation precision across each pair of classes in Figure 3. We see that classes 0.2 and 0.8, who have the biggest gap, are separated first (Figure 3(b)), then 0.4 and 0.8 (Figure 3(c)) and finally 0.2 and 0.4 (Figure 3(a)).

Finally, for mean estimation, we provide the per-class counterparts of Table 2 in Tables 3-5. In line with previous results on class estimation, agents of class 0.8 are the ones that converge faster to the desired mean estimation accuracy. Interestingly, observe that for $\epsilon = 0.1$, agents of class 0.2 converge almost as fast as those from class 0.8: this is because they do not need to eliminate all agents from class 0.4 to reach this accuracy. This illustrates how our approach can naturally adapt to the gaps and desired estimation accuracy.

### F.2 Results on a 2-class problem

We experiment with a 2-class problem generated in the same way as the 3-class problem considered in the main text, except that the means are chosen among $\{0.2, 0.8\}$. This makes the problem easier since the gap between the two classes corresponds to the largest gap in the 3-class problem. The results shown in Figure 4 reflect this: agents correctly identify their class and reach highly accurate mean estimates much faster than

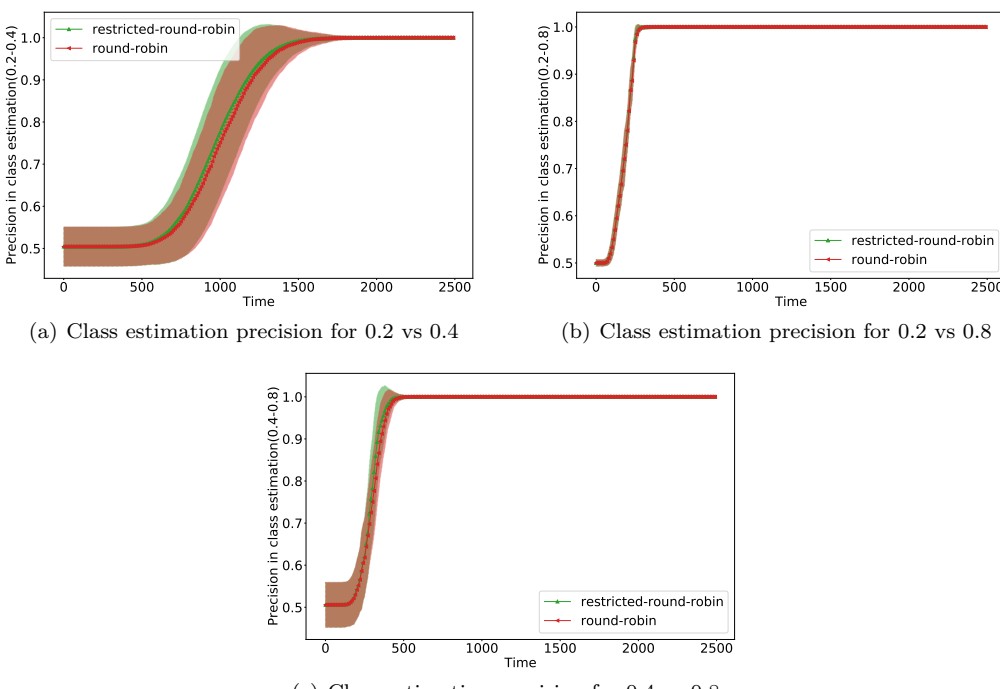

(a) Class estimation precision for 0.2 vs 0.4

(b) Class estimation precision for 0.2 vs 0.8

(c) Class estimation precision for 0.4 vs 0.8

Figure 3: Class estimation precision across time for each pair of classes on the 3-class problem.

| Algorithm | conv(0.1) | | conv(0.01) | |
|---|---|---|---|---|
| | avg/std | max | avg/std | max |
| Round-Robin (RR) | $289 \pm 92$ | 594 | $1207 \pm 170$ | 1782 |
| Restricted-RR | $271 \pm 77$ | 568 | $1194 \pm 177$ | 1857 |
| Soft-Restricted-RR | $111 \pm 30$ | 239 | $761 \pm 173$ | 1379 |
| Aggressive-Restricted-RR | $74 \pm 40$ | **215** | $\mathbf{382 \pm 95}$ | **830** |
| Local | $\mathbf{41 \pm 40}$ | 288 | $4548 \pm 4075$ | 28616 |
| Oracle | *6 ± 4* | *15* | *100 ± 63* | *246* |
| Theoretical Restricted-RR $(\tau_a)$ | 3878 | | 33406 | |
| Theoretical Local $(\tau_a)$ | 885 | | 100216 | |

Table 3: Empirical convergence times (see Eq. 12) for class 0.2 of different algorithms on the 3-class problem (Gaussian distributions with true means 0.2, 0.4, 0.8) for a target estimation error of $\epsilon = 0.1$ (unfavorable regime) and $\epsilon = 0.01$ (favorable regime). We report the average, standard deviation and maximum across agents and runs. We also report the high-probability mean estimation times $\tau_a$ given by our theory for *Restricted-Round-Robin* and *Local*.

in the 3-class problem. Consequently, the improvement compared to *Local* is even more significant and our approach almost matches the performance of *Oracle*. We omit the per-class figures as they are essentially the same as Figure 4(b).

| Algorithm | conv(0.1) | | conv(0.01) | |
|---|---|---|---|---|
| | avg/std | max | avg/std | max |
| Round-Robin (RR) | $721 \pm 160$ | 1239 | $1232 \pm 181$ | 2088 |
| Restricted-RR | $708 \pm 158$ | 1175 | $1216 \pm 183$ | 1925 |
| Soft-Restricted-RR | $\mathbf{31 \pm 33}$ | 340 | $829 \pm 188$ | 1432 |
| Aggressive-Restricted-RR | $35 \pm 42$ | 340 | $\mathbf{420 \pm 92}$ | **832** |
| Local | $41 \pm 39$ | **277** | $4482 \pm 3846$ | 26488 |
| Oracle | $4 \pm 3$ | $19$ | $99 \pm 70$ | $303$ |
| Theoretical Restricted-RR ($\tau_a$) | 3878 | | 33406 | |
| Theoretical Local ($\tau_a$) | 885 | | 100216 | |

Table 4: Empirical convergence times (see Eq. 12) for class 0.4 of different algorithms on the 3-class problem (Gaussian distributions with true means 0.2, 0.4, 0.8) for a target estimation error of $\epsilon = 0.1$ (unfavorable regime) and $\epsilon = 0.01$ (favorable regime). We report the average, standard deviation and maximum across agents and runs. We also report the high-probability mean estimation times $\tau_a$ given by our theory for *Restricted-Round-Robin* and *Local*.

| Algorithm | conv(0.1) | | conv(0.01) | |
|---|---|---|---|---|
| | avg/std | max | avg/std | max |
| Round-Robin (RR) | $271 \pm 35$ | 401 | $379 \pm 38$ | 538 |
| Restricted-RR | $266 \pm 31.25$ | 384 | $342 \pm 38$ | **526** |
| Soft-Restricted-RR | $100 \pm 36$ | 232 | $280 \pm 78$ | 958 |
| Aggressive-Restricted-RR | $57 \pm 24$ | **172** | $\mathbf{222 \pm 90}$ | 958 |
| Local | $\mathbf{41 \pm 39}$ | 289 | $4434 \pm 3883$ | 27470 |
| Oracle | $6 \pm 4$ | $17$ | $95 \pm 56$ | $208$ |
| Theoretical Restricted-RR ($\tau_a$) | 1085 | | 33406 | |
| Theoretical Local ($\tau_a$) | 885 | | 100216 | |

Table 5: Empirical convergence times (see Eq. 12) for class 0.8 of different algorithms on the 3-class problem (Gaussian distributions with true means 0.2, 0.4, 0.8) for a target estimation error of $\epsilon = 0.1$ (unfavorable regime) and $\epsilon = 0.01$ (favorable regime). We report the average, standard deviation and maximum across agents and runs. We also report the high-probability mean estimation times $\tau_a$ given by our theory for *Restricted-Round-Robin* and *Local*.

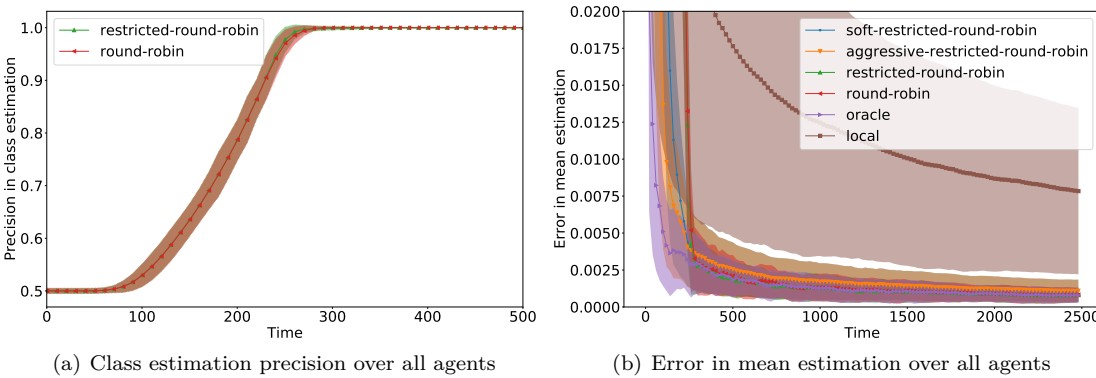

(a) Class estimation precision over all agents

(b) Error in mean estimation over all agents

Figure 4: Results for the 2-class problem (Gaussian distributions with true means 0.2 and 0.8).

