# OpenReview forum: "Collaborative Algorithms for Online Personalized Mean Estimation"
_TMLR — Accepted by TMLR_

### Review · Reviewer_VN18 · 2022-09-20

**Summary Of Contributions:**

This paper proposes collaborative algorithms for personalized mean estimation in the online setting. Specifically, each user draws samples from a subgaussian distribution with the same parameter but potentially different means and aims to estimate their mean using data from their distribution but also leveraging data from other users who have the same (or close) mean. In each round, each user draws a sample from their distribution to update their local mean estimate, and then are allowed to query one user (to be chosen according to some policy) for their current local mean estimate and store it. Each user’s estimate for the round is a weighted average of their local mean and the local means of other users that they have queried in the past rounds. This paper explores two policies for choosing which user to query in each round and several policies for choosing the weighting scheme for the mean estimate of the round.

The paper provides a theoretical analysis for a certain choice of policies, which shows that for a given user, after a number of rounds that roughly corresponds to the number of samples required to distinguish between the user’s distribution from the one with the closest mean,  the user has a good estimate of their mean with high probability and they have identified the users that belong in the same class (i.e., share the same mean). This number of rounds, under some natural parameter regimes, is smaller than the time it would take for the user to estimate their mean on their own, by roughly the size of the class they belong to.

An extension of this result to relax the assumption of users from the same class sharing exactly the same mean is included. Finally, the paper shows empirically how these policies perform for simulated data.

**Broader Impact Concerns:**

This work does not have obvious ethical implications.

**Requested Changes:**

The following is not crucial for my recommendation, but I would suggest motivating the setting more or adding a longer comparison with other settings of collaborative learning and distributed clustering. I actually think that phrasing the problem in terms of sample complexity and not convergence time might make the contributions easier to understand, but as I mentioned, motivating the online view might be equally satisfying.

A small typo in page 7: “identity the class”->identify the class

**Strengths And Weaknesses:**

### Strengths ###
- Personalized mean estimation is a problem that sees increasing interest in the ML community. This paper studies a version of this problem.
- The authors consider and compare empirically several policies to instantiate their framework, which collects local mean estimates over time and each round’s estimate is their weighted average.
- The proofs seem to be correct and complete.

### Weaknesses: ###
- I find the use of the online setting here a bit odd. There is no notion of regret — the error that is being optimized is the absolute distance between the estimate of a given user and their true mean. The number of rounds in the analysis is only used as a proxy to the number of samples that the users have drawn. As long as the number of samples is high enough, their local estimates are accurate enough so that we can distinguish who is in the same class and use everyone’s samples for the final estimate. So it confused me that the problem is presented as an online learning problem with the updates being restricted to follow this pattern. I don’t particularly object to this version of the problem, but my concern is that if we remove the constraint that each user receives one sample at a time and they are allowed to query one other user at a time, then there are a lot of other settings and works to compare to. For example, if we are interested in personalized mean estimation in the centralized setting (everyone pools their data together) there are several works in the multitask learning literature to consider. Other work that seems relevant is in distributed clustering (with communication constraints even). I think it is fine to say that these are not related because the problem in this specific online form, but then I would expect the setting to be more motivated.
- The techniques are very similar to general bandits where convergence depends on the confidence intervals and their overlap. This is a minor comment however, since a highly valuable technical contribution is not necessary.

---

> ### Author Response · Authors · 2022-10-25
> **Response to reviewer VN18**
>
> We thank the reviewer for the feedback.
>
> - *On the online aspects of our problem and approach.* Thank you for the comment. We emphasize that many online problems are not about minimizing a notion of regret but instead consider a *pure exploration* setting. This is notably the case of best arm identification in multi-armed bandits (Audibert et al., 2010), where the goal is to identify the best arm as fast as possible. In our approach, one subproblem is to identify the correct class as fast as possible, and we do so by discarding far-away agents in an online way so they can stop being queried as early as possible, much like in approaches for best arm identification. Similarly, our global objective is to reach an $\epsilon$-approximation of the personal mean as fast as possible. It internally relies on the online estimation of the class, as well as on some weighting schemes which can adjust the influence of agents based on the confidence intervals estimated online. For all these reasons, we believe that the term online is appropriate to describe our problem setting and our approach.
>
>     We have added a few sentences before Definition 1 to emphasize that we consider a pure exploration setting. In Section 2, we have also highlighted  that some related work like Réda et al. (2022) also consider a pure exploration setting. Please let us know if there is something more we can do.
>
> - *Typo.* Thanks for spotting the typo, we have corrected it.

---

### Review · Reviewer_v3Wz · 2022-09-23

**Summary Of Contributions:**

This paper studies the problem of online collaborative personalized mean estimation, motivated federated learning systems. A particular problem formulation is considered where, at each round, each agent receives one random observation of the random variable centered at the parameter they wish to estimate, and they may also query the running-estimate mean of one other agent. A few strategies are studied, including round-robin and restricted round-robin for determining which other agent to query next, and three possibilities for how to aggregate and fuse information from other agents with the local observations to ultimately form an estimate of the mean of each agent. Theory is provided for two cases, one where agents are partitioned into equivalence classes and those agents within the same class have exactly the same mean, and a more relaxed case where agents simply aim to fuse their observations with those of other agents whose means are similar enough to their own. Experiments are also provided to illustrate the theory for the first case.

**Broader Impact Concerns:**

None noted.

**Requested Changes:**

1. The prior work of [Cheng et al. (2021)](https://arxiv.org/abs/2108.07313v3) also aims to formalize a related personalization problem, including characterizing bias-variance tradeoffs. While the formulation is certainly different, it would be good to cite and discuss the relationship to this work.

2. In cross-device federated learning, typically agents (devices) communicate with a server, rather than communicating directly with each other, since device availability is sporadic. Could the proposed formulation and methods be adapted to this setting in a natural way? This could also be discussed as a possible direction for future work.

3. This is not necessarily a requested change, but I find it surprising that round-robin and restricted round-robin perform so similarly in Figure 1. Is this expected? Wouldn't one expect restricted round-robin to be much better than round-robin? Please discuss.

4. Section 6.2 currently states that
> We can see that the classes 0.2 and 0.8 are separated very early, quickly followed by 0.4 and 0.8 and finally, after sufficiently many samples have been collected, the pair with the smallest gap (0.4 and 0.2).

Although we can see points where precision increases, there is no evidence in Fig 1(a) to directly support the claims about which classes are being separated (even though the statement does align with what one would intuitively expect). Can additional evidence be provided to support the claim?

5. For finite $A$, in the limit as $\eta \rightarrow 0$, the results in Section 7 should reduce to those in Section 5. Is that correct? It's not obvious to me that Theorem 4 reduces to that of Theorem 2, however (in particular, when there are two or more agents with exactly the same mean). Please discuss.

6. By adopting the approach of optimism in the face of uncertainty, the results in the paper depend on first exactly estimating the equivalence class. In this case, the improvement in terms of $|\mathcal{C}_a|$ is evident. In practice, if agent $a$ can carefully rule out those agents which are not in its equivalence class, so that $\mathcal{C}_a^t$ is strictly a subset of $\mathcal{C}_a$, then couldn't it still expect some improvement in its mean estimate, albeit not the greatest reduction in variance achievable? It would be nice to see some further discussion of this tradeoff, in particular, with an eye towards the large-scale setting, where one might hope to form accurate estimates without needing to communicate with every other agent in the system.

7. Although the writing is clear for the most part, there are some parts of the paper that are awkward to read. The first sentence of the introduction is one example, where "wide spreading" should probably be "widespread" and "pregnant" seems unlikely to be the word that was intended here (maybe "pressing" or "urgent"?).

**Strengths And Weaknesses:**

## Strengths
1. The problem of personalized learning and estimation is timely and highly relevant. While several approaches have been proposed in the literature and evaluated empirically, relatively little work has taken a more formal and rigorous approach as in this paper.
2. The paper is generally well-written (although see below for some minor suggestions/typos).
3. The theoretical results appear to be correct and verify the intuition of what one might hope for in the particular model considered.

## Weaknesses
1. The details of the particular problem formulation proposed are not well-motivated, and several aspects seem disconnected from practice.
* The paper uses federated learning to motivate the problem setting. One key tenet of federated learning is to preserve the privacy of each individual agent's data. The proposed setting and methods involve agents sharing their observations directly with other agents, which directly violates this tenet.
* In the proposed formulation, each agent may request the data of only one other agent at each round. This assumption pins a particular communication-computation tradeoff (one observation = one communication). In distributed/decentralized estimation problems it is often useful to parameterize the formulation in such a way that one can explore different tradeoff regimes (e.g., if communication is either much more costly or much less costly than gathering new observations).


2. The proposed approach is not scalable for several reasons. Primarily, it is assumed that the amount of memory at each agent is proportional to the total number of agents $A$. This is impractical in most large-scale systems.

I don't think that either of the above are fatal weaknesses, given that this is an area which has not been widely studied and that the paper contains many useful insights. But it is worth acknowledging and discussing these, along with ways they might be addressed in future work.

3. The experimental evaluation illustrates some of the ideas from the paper, but it could be improved. In my view the main contributions of the paper are the problem formulation, proposed approaches, and theoretical guarantees. The experiments should then serve to validate the theory. While the experiments illustrate trends that the theory suggests, they don't actually include a curve for each of the bounds provided in Theorems 1 and 2. It would be useful to include these to get a sense of how loose/tight the theory is. Since no theoretical guarantees are provided for the soft and aggressive heuristics, experiments could also be used to better illustrate their behavior. However, the current experiments focus on one particular setting (3 agents, with only one specific choice of means; and a two-agent example in the appendix), so they do not help identify extremes or edge cases, to understand if or when the heuristics may fail (gracefully, or not), or if they have any limitations. Also, there are no experiments provided for the more interesting/relevant setting of non-separated classes considered in Section 7.

---

> ### Author Response · Authors · 2022-10-25
> **Response to reviewer v3Wz (1/2)**
>
> We thank the reviewer for the numerous insightful comments. We respond to each point below.
>
> ### Weaknesses
>
> 1. We adress the two points (privacy and communication) below.
>     - As in federated learning, agents only share aggregate summaries of data (here, local averages), not individual observations. That being said, due to the online nature of data collection, in the first rounds these summaries are indeed composed of a small number of observations (a single one at $t=1$), which may raise privacy issues when data is sensitive. A simple solution to address this concern is for agents to collect a certain number of observations before they start collaborating (or even before each communication, see next point).
>     - The setting of "one observation = one communication" is standard: virtually all the collaborative multi-armed bandits approaches discussed in Section 2 consider this. Nevertheless, we agree with the reviewer that it can be relevant to consider different trade-offs between communication and data collection. We note that it is straightforward to adapt our approach to the case where each agent collects $m$ samples between each communication (to reduce communication and/or improve privacy, see previous point). This amounts to multiplying by $m$ the number of observations in our confidence intervals and empirical estimates. Extensions to cases where the number of samples between each communication is random and/or varies across agents are interesting directions for future work.
>
>     We have added the above discussions in Remark 2 (Section 5).
>
> 2. In large-scale systems, it may indeed be impractical for agents to query all other agents and/or to maintain a memory size that is linear in the number of agents $A$. From a practical point of view, each agent can instead consider a restricted subset of agents of reasonable size. This subset could be picked uniformly at random, be composed of neighoring nodes (in the network or physical world), or (when available) be based on prior knowledge on who is more likely to be in the same class. We have added this as Remark 1 in Section 3.
>
>     An interesting direction to allow more exploration in large-scale systems could rely on the idea of peer sampling [1], i.e., randomly sampling a few agents from time to time to discover potential new members of the class beyond the initial subset. This is now mentioned as future work in Section 8.
>
>     [1] Jelasity, M., Voulgaris, S., Guerraoui, R., Kermarrec, A.-M., and van Steen, M. (2007). Gossip-based peer sampling. ACM Trans. Comput. Syst., 25(3).
>
> 3. First, let us point out that our experiments consider $A=200$ agents, not 3 or 2 (this corresponds to the number of classes).
>
>     Second, we thank the reviewer for the interesting suggestions to improve our experimental section. To accommodate them, we have revised our experimental section to provide more information, including:
>     - For mean estimation, in addition to the average and standard deviation across agents and runs, we have also provided in Table 2 the maximum time as well as the theoretical high-probability times $\tau_a$ given by our theory for *Restricted-Round-Robin* and *Local*. Comparing the maximum empirical times and the theoretical high-probability times gives an idea of the tightness of the analysis. We also provide per-class tables in the appendix.
>     - We have also added Table 1, which provides similar statistics for class estimation, including the theoretical value $\zeta_a$ given by our analysis.
>
>     These changes allow to get a sense of the tightness of our theoretical results, as requested by the reviewer. We did not have the time to run more experiments with more values of $\epsilon$ or a different number of agents, but we can provide such additional experiments for the final version if the reviewer thinks this is needed.
>
>     Finally, we want to point our that the reviewer's feedback helped us catch two errors in our experimental results, which were to the disadvantage of our approach and that we have fixed in the revised version of the paper. First, we were mistakenly using larger confidence sets than given by our analysis, resulting in larger times for class identification and in turn larger mean estimation times for all variants of our approach. Second, we realized that most of the runs for *Local* did not converge to accuracy $\epsilon=0.01$ under time horizon $2500$, which made the results of *Local* look better as the convergence time was capped at 2500. We now use a time horizon of $30000$ for *Local*. We would like to thank the reviewer again for his/her feedback.

---

> > ### Author Response · Authors · 2022-10-25
> > **Response to reviewer v3Wz (2/2)**
> >
> > ### Requested changes
> >
> > 1. Thank you for the reference to Cheng et al. (2021). The authors consider the batch learning setting, so we have added this reference at the end of the first paragraph of the related work section where we discuss such methods. We emphasize there that it is one of the only work that makes clear statistical assumptions and provides error bounds with respect to the underlying distributions.
> >
> > 2. We first emphasize that it is increasingly common to consider peer-to-peer communications in cross-device FL as a way to improve scalability (see Kairouz et al. 2021, Section 2.1 therein). In our case, we think it is more natural to present things in this way, but it is possible to implement our approach by communication through a server (which is often how peer-to-peer communication is implemented in practical systems). For simplicity, if we consider the case that everyone can message the server and it can message back to everyone, the extension of our work to this setting is straightforward: everyone updates the server first and each agent queries the agent wanted and asks the value from the server. We believe that investigating more complex scenarios (e.g., where the number of message is limited or when one seeks to minimize the number of communication rounds) is interesting but out of the scope of this paper.
> >
> > 3. As shown by our theoretical results (Theorem 1), the difference between round-robin and restricted-round-robin for class estimation time is the term $\sum_{l \in [A]\backslash C_{a}} 1_{n^*_{a,a} > n^*_{a, l}+A - 1}$. This corresponds to the number of agents that are already removed from the set of similar agents when we reach time $n^*_{a,a}$ (the time needed to eliminate the agent with smallest nonzero gap). Then, for vanilla round-robin, we should still go through $A - 1$ other agents, while for restricted-round-robin we only need to check the $A - 1 - \sum_{l \in [A]\backslash C_{a}} 1_{n^*_{a,a} > n^*_{a, l}+A - 1}$ remaining agents. This rather small gain is due to the problem setting: recall that querying an agent at time $t$ yields the full statistics of observations of this agent up to time $t$. We have clarified this in Section 6.2.
> >
> > 4. Thanks for pointing the ambiguity in Figure 1(a). We have added plots to the appendix (Figure 3) to support our claim, and refer this plot when we make our observation in the main text.
> >
> > 5. For the setting of imperfect classes (Section 7), the objective is different: we aim to estimate the mean of the class (see Eq. 13). We thus adapt the weighting scheme accordingly by giving uniform weights to the local estimates of each member of the (optimistic) class, see discussion before Theorem 3. This is different from giving each sample the same weight as done for the perfect class case. This explains the difference you noticed. We have expanded the discussion after Theorem 4 to make this more clear.
> >
> > 6. This is an inherent limitation of the optimistic approach, where we consider another agent as similar as long as it remains sufficiently plausible that it is indeed in the same class. As the reviewer notes, this means we can only guarantee a gain once the class has been correctly identified. The heuristic weighting schemes we introduce allow to mitigate this effect by reducing the importance of some agents before there is enough evidence to formally eliminate them from the class. In the large-scale setting, we can also restrict the notion of class to a subset of agents of reasonable size to identify the class faster (see our response above).
> >
> >     An alternative strategy would be to act conservatively, where other agents are considered to be different until we are sufficiently confident that they belong to the same class. The downside of this approach is that agents would need the knowledge of gaps in order to determine when an agent can be considered to be in the same class. This is a big limitation in practice. In contrast, our approach does not require such knowledge. Note that we discuss this point in the conclusion of the paper (we have now highlighted this more by using a titled paragraph).
> >
> > 7. Thank you for spotting these issues, we have corrected them.

---

> > > ### Comment · Reviewer_v3Wz · 2022-10-26
> > > **Thank you**
> > >
> > > Thanks for the changes you made and for the responses to points raised in my review. Indeed, I was confusing the number of agents and the number of classes used in the experiments; thanks for clarifying this.
> > >
> > > I am satisfied with the other changes that have been made.

---

> > ### Comment · Reviewer_v3Wz · 2022-10-26
> > **Satisfied with experiments already added**
> >
> > Thank you for your responses and the changes you have made in response to the points I raised.
> >
> > I agree that the inclusion of Tables 1 and 2 already help to paint a clearer picture of the relationship between the theoretical bounds and empirical observations. Although additional experiments would further strengthen the paper, I'm satisfied with the current state and will not insist on more.
> >
> > That there were some errors in your previous experiments is understandable, and at the same time does also bring to question how much confidence to put in the empirical evaluation. Will code to reproduce your results also be released along with the paper? I strongly encourage this.
> >
> > Regarding the first weakness (privacy), the point I wanted to bring up is not just about the number of observations at each device. Sharing aggregated observations may still reveal private information, e.g., if the distribution of observations is correlated with a characteristic the device wishes to keep private. Future work may involve incorporating differential privacy to overcome this limitation. Alternatively, in a system involving a central server, the server may aggregate responses from several clients.

---

> > > ### Author Response · Authors · 2022-10-28
> > > **Thank you - updated code and added privacy in future work**
> > >
> > > We thank the reviewer for the feedback on our response and revision. We are happy to see that the reviewer is generally satisfied.
> > >
> > > Regarding the code to reproduce the experiments, we will indeed release it along with the paper. Note that we had attached the code as supplementary material to the original submission, and we have now updated it and simplified it a bit.
> > >
> > > Regarding privacy, we totally agree that aggregate statistics can still leak sensitive information. We have added a paragraph to the conclusion on providing formal differential privacy guarantees and analyzing the resulting privacy-utility trade-offs for the problem we consider.
> > >
> > > We thank again the reviewer for his/her feedback.

---

### Review · Reviewer_2AwD · 2022-10-10

**Summary Of Contributions:**

This paper studies the mean estimation problem in an online fashion where multiple agents with the same or different means could collaborate with each other to improve their own estimation by incorporating others’ information. The authors propose an algorithm where each agent can use other agents’ information in an optimistic way by estimating the confidence interval of their estimation. The agent will choose another agent to query in a round robin way or restricted round robin way (only choose agents that are estimated to have the same mean). The authors provide a time complexity analysis of their algorithm when using a simple weighting scheme to sum the information within one group with the same mean. They also conducted experiments to show the performance of the algorithm with respect to the time complexity.


**Requested Changes:**

I think more discussion on the motivation of the multi-agent setting, detailed instance-dependent complexity, and comparison with other methods are necessary to fully evaluate the contribution of the submission. A more comprehensive experimental study on the different weighting schemes and the querying strategies proposed in this paper would improve and strengthen the work.

**Strengths And Weaknesses:**

This paper is well written especially for the algorithm and analysis sections. The algorithm idea is articulated clearly and the theoretical analysis looks good to me. However, there are some aspects that need more improvement such as the motivation of the problem, the comparison of the complexity with other methods, and the trade-off between the communication and time complexity.

I am confused about the motivation of querying other agents’ estimates just in order to estimate the mean well. Since the goal of each agent $a$ is just to estimate the mean $\mu_a$ of distribution $\nu_a$, what is the benefit/motivation for agent a to query the mean of other agents? I might be missing something here. But the time complexity of the proposed method is not better than that of a single agent algorithm who just uses $O(1/\epsilon^2)$ data to concentrate its empirical mean to the true mean up to an error $\epsilon$.

If the goal is to estimate the mean in a distributed way, then the authors should comment and discuss more about the communication complexity and memory complexity of the proposed algorithm. For example, the communication complexity seems to be linear in the iteration number since the algorithm requires to ask another agent’s estimation at every iteration. Is the memory complexity $|A|^2$? In this case, I do not see the benefit of such a collaborative algorithm compared with aggregating all the data for one agent on the same server, since the time complexity and the communication complexity are the same.

Although based on a rough calculation it seems to me that the time complexity in Theorem 2 is in the order of $O(1/\Delta_a^2+1/\epsilon^2)$ plus additional terms, it would be much better if the authors can provide a more specific instance dependent result in the theorem or the discussion, which will makes it much easier to interpret and compare the result.

The theoretical analyses are based on the simple weight scheme. However, according to the experimental results, it seems that the performance of the algorithm with the simple weight scheme is significantly worse than the other two heuristic weighting schemes proposed just for the experiments. This discrepancy between the theory and the empirical results makes the conclusion unclear whether the proposed algorithm works in more complicated real-world applications. The authors might need to conduct more experiments on different problems and with ablation study to show when these different schemes work in practice.

---

> ### Author Response · Authors · 2022-10-25
> **Response to reviewer 2AwD**
>
> We thank the reviewer for the insightful feedback.
>
> - *Clarification on time complexity and improvements compared to local estimation.* There is some confusion regarding the theoretical time complexity of our approach and the improvements compared to local estimation. The reviewer is correct is saying that $\widetilde{O}(1/\epsilon^2)$ samples are needed to guarantee $\epsilon$-accuracy. For local estimation (i.e., single agent setting), this requires $\widetilde{O}(1/\epsilon^2)$ time since the agent only relies on its local samples. Our theoretical results show that our approach (Restricted Round-Robin with simple weighting) achieves $\epsilon$-accuracy in $\widetilde{O}
> (\max\{1/\Delta_a^2,1/\epsilon^2|\mathcal{C}_a|\})$ time for an agent in class $\mathcal{C}_a$, where $\Delta_a$ is the gap between class $\mathcal{C}_a$ and the others, and $|\mathcal{C}_a|$ is the number of agents in $\mathcal{C}_a$. This is much better than the $O(1/\Delta_a^2+1/\epsilon^2)$ claimed by the reviewer. In particular, our approach is **faster than local estimation** (i.e., single agent setting) as soon as the time $\zeta_a$ needed to correctly identify $\mathcal{C}_a$, which is of order $\widetilde{O}
> (1/\Delta_a^2)$, is smaller than the time needed for local estimation to reach $\epsilon$-accuracy, which is of order $\widetilde{O}(1/\epsilon^2)$. Furthermore, for large enough gaps or small enough $\epsilon$, **our approach can be up to $|\mathcal{C}_a|$ times faster than local estimation, which is nearly optimal** as this corresponds to the time complexity of the *Oracle* baseline (which is given oracle knowledge of the classes).
>
>     We hope that the above discussion lifts the doubts of the reviewer about our theoretical guarantees and convinces him/her of their significance for the multi-agent setting. We acknowledge that the discussion after Theorem 2 was perhaps not clear enough, so we have tried to improve it and we added the main orders of magnitude in $\widetilde{O}(\cdot)$ notation. Finally, we also emphasize that the practical behavior of our approach is in line with the above theory, as we illustrate more clearly in the revised experimental section.
>
>
> - *Comparison of the complexity with other methods*. We are not aware of other methods we could compare to beyond the *Local* and *Oracle* baselines. We think that the main contributions of our paper is to introduce this new problem, to propose an algorithm for it, and to theoretically analyze it.
>
> - *Reducing the communication complexity.* The communication complexity is indeed linear in the number of time steps. As explained in our response to reviewer v3Wz, a simple way to reduce communication is to consider the case where each agent collects $m$ samples between each communication. This adaptation is straightforward, as it amounts to multiplying by $m$ the number of observations in our confidence intervals and empirical estimates. We have mentioned this in Remark 2 in Section 5.
>
> - *Reducing the memory complexity.* The memory complexity at each agent is indeed linear in the number of agents, which may be impractical in large-scale systems. However, as explained in our response to reviewer v3Wz, in practice each agent can instead consider a restricted subset of agents of reasonable size. This subset could be picked uniformly at random, be composed of neighoring nodes (in the network or physical world), or (when available) be based on prior knowledge on who is more likely to be in the same class. We have added this as Remark 1 in Section 3.
>
> - *Better empirical evaluation of weighting schemes.* We believe our revised results give a good overview of the performance of the different weighting schemes. Note that we already do a kind of ablation study by considering all variants of our approach (with the different weighting schemes). If the reviewer has a concrete suggestion to improve the empirical evaluation on this aspect, we would be happy to consider it.

---

> ### Author Response · Authors · 2022-10-28
> **Clarification on the problem setting**
>
> Upon reading again the review of reviewer 2AwD, in complement to our initial response, we would like to explicitly address the following question asked by the reviewer:
>
> > I am confused about the motivation of querying other agents’ estimates just in order to estimate the mean well. Since the goal of each agent $a$ is just to estimate the mean of distribution $\nu_a$, what is the benefit/motivation for agent a to query the mean of other agents? I might be missing something here.
>
> The reason why collaboration is beneficial in our personalized mean estimation problem is that we assume that **several agents have the same (or similar) mean**, thereby defining classes of agents (Definition 2, or Definition 7 for the imperfect class case). The challenge is that **the number of classes and the class membership of agents are unknown and must be discovered online**. Our approach allows each agent to identify its class in an online way by querying other agents, with guarantees on the time needed to identify the class correctly based on the mean gap with the other classes (Theorem 1). This in turn allows us to provide better mean estimation time complexity for our approach than for local estimation (single agent setting) when identifying the class is faster than locally estimating the mean to the desired precision (Theorem 2). We refer to our previous response for details on the time complexities and improvements compared to local estimation.

---

### Author Response · Authors · 2022-10-21
**Thank you for your reviews - response and revision will be submitted soon**

Dear reviewers,

Thanks a lot for your feedback and the time spent on our work. We are preparing a detailed response as well as a revision of the paper to account for your comments, which we will submit early next week. We look forward to the subsequent discussion with you!

Best regards,
The authors

---

### Author Response · Authors · 2022-10-25
**Responses and revised version of the paper**

We thank the reviewers for their time and insightful comments. We have provided a detailed response to each review, and we have done our best to follow the recommendations of the reviewers to prepare a revised version of the paper within the allocated time. We summarize these changes below (all of them are highlighted in blue color in the revised version of the manuscript).

- We updated the related work and the setting sections to better position the paper with respect to online learning and the fact that we consider a pure exploration setting, similar to best arm identification for multi-armed bandits. We have also added a reference to Cheng et al. (2021).
- We added Remark 1 to discuss the scalability issues and propose simple ways to address the large-scale setting. We also discuss this further as part of the future work in Section 8.
- We added Remark 2 to discuss possible extensions to the setting and the algorithms that can reduce communication and mitigate privacy concerns. In particular, the restriction that agents communicate each time they receive one sample can be easily relaxed.
- We have clarified the discussion of the theoretical time complexities for class and mean estimation, giving the main orders of magnitude for our approach and better highlighting the gains with respect to local estimation (single agent setting).
- Finally, we updated the experimental section in the main text and in the appendix, thanks to the insighful remarks of the reviewers. In particular, we compare the theoretical values given by our analysis to the empirical ones.

We hope that the reviewers will find our responses and revised version convincing. We are happy to continue the discussion if there are additional points to address.

---

### Decision · Action_Editors · 2022-12-12

**Recommendation:** Accept as is

**Comment:**

This paper studies the problem of online collaborative personalized mean estimation. The problem formulation considered here is that at each round, each agent receives one random observation of the parameter they wish to estimate, and can also query the running-estimate mean of one other agent. The paper studies different strategies including round-robin and restricted round-robin for determining which other agent to query next, and three possibilities for how to aggregate and fuse information from other agents with the local observations to ultimately form an estimate of the mean of each agent.

Overall the relevance of the problem studied by the paper was appreciated by the reviewers. The paper's results were deemed to be correct, accurate and useful by the reviewers. The reviewers' requests were also incorporated by the authors. Overall I am happy recommending an accept which was the unanimous recommendation by the reviewers.


**Audience:**

Yes. Researchers in federated learning and online learning will be interested in this paper.

**Claims And Evidence:**

The claims made by the paper are accurate and convincing. The questions that the reviewers asked have been adequately addressed.

---

> ### Author Response · Authors · 2022-12-19
> **Camera-ready version submitted**
>
> Dear Action Editor,
>
> Thank you for handling our submission! We have submitted the camera-ready version.
>
> Best
> The Authors